



# Drivers for spatial, temporal and long-term trends in atmospheric ammonia and ammonium in the UK

Yuk S. Tang[1], Christine F. Braban[1], Ulrike Dragosits[1], Anthony J. Dore[1], Ivan Simmons[1], Netty van Dijk[1],
Janet Poskitt[2], Gloria Dos Santos Pereira[2], Patrick O. Keenan[2], Christopher Conolly[3], Keith Vincent[3],
Rognvald I. Smith[1], Mathew R. Heal[4] & Mark A. Sutton[1]

[1]CEH, Bush Estate, Penicuik, Midlothian EH26 0QB
[2]CEH, Lancaster Environment Centre, Bailrigg, Lancaster LA1 4AP
[3]Ricardo Energy & Environment, Gemini Building, Fermi Avenue, Harwell, Oxon OX11 0QR
[4]School of Chemistry, University of Edinburgh, David Brewster Road, Edinburgh EH9 3FJ

*Correspondence to*: Y. Sim Tang (yst@ceh.ac.uk)

**Abstract.** A unique long-term dataset from the UK National Ammonia Monitoring Network (NAMN) is used here to assess
spatial, seasonal and long-term variability in atmospheric ammonia ($NH_3$: 1998-2014) and particulate ammonium ($NH_4^+$: 1999-
2014) across the UK. Extensive spatial heterogeneity in $NH_3$ concentrations is observed, with lowest annual mean
concentrations at remote sites ($< 0.2 \ \mu g \ m^{-3}$) and highest in the areas with intensive agriculture (up to 22 $\mu g \ m^{-3}$), while $NH_4^+$
concentrations show less spatial variability (e.g. range of 0.14 to 1.8 $\mu g \ m^{-3}$ annual mean in 2005). Temporally, $NH_3$
concentrations are influenced by environmental conditions and local emission sources. In particular, peak $NH_3$ concentrations
are observed in summer at background sites (defined by 5 km grid average $NH_3$ emissions <1 kg N ha$^{-1}$ y$^{-1}$) and in areas
dominated by sheep farming, driven by increased volatilization of $NH_3$ in warmer summer temperatures. In areas where cattle,
pig and poultry farming is dominant, the largest $NH_3$ concentrations are in spring and autumn, matching periods of manure
application to fields. By contrast, peak concentrations of $NH_4^+$ aerosol occur in spring, associated with long-range
transboundary sources. An estimated decrease in $NH_3$ emissions by 16 % between 1998 and 2014 was reported by the UK
National Atmospheric Emissions Inventory. Annually averaged $NH_3$ data from NAMN sites operational over the same period
($n = 59$) show an indicative downward trend, although the reduction in $NH_3$ concentrations is smaller and non-significant (−6.3
% (Mann-Kendall, MK); −3.1 % (linear regression, LR)). In areas dominated by pig and poultry farming, a significant
reduction in $NH_3$ concentrations between 1998 and 2014 (−22 % (MK); −21 % (LR): annually averaged $NH_3$) is consistent
with, but not as large as the decrease in estimated $NH_3$ emissions from this sector over the same period (−39 %). By contrast,
in cattle-dominated areas there is a slight upward trend (non-significant) in $NH_3$ concentrations (+ 12%, (MK); + 3.6% (LR):
annually averaged $NH_3$), despite the estimated decline in $NH_3$ emissions from this sector since 1998 (−11%). At background
and sheep dominated sites, $NH_3$ concentrations increased over the monitoring period. These increases (non-significant) at
background (+17 % (MK); +13 % (LR): annually averaged data) and sheep dominated sites (+15 % (MK); +19 % (LR):
annually averaged data) would be consistent with the concomitant reduction in $SO_2$ emissions over the same period, leading
to a longer atmospheric lifetime of $NH_3$, thereby increasing $NH_3$ concentrations in remote areas. The observations for $NH_3$
concentrations not decreasing as fast as estimated emission trends are consistent with a larger downward trend in annual
particulate $NH_4^+$ concentrations (1999-2014: −47 % (MK); −49 % (LR), $p < 0.01$, $n = 23$), associated with a slower formation
of particulate $NH_4^+$ in the atmosphere from gas-phase $NH_3$.



## 1    Introduction

Atmospheric ammonia ($NH_3$) gas is assuming increasing importance in the global pollution climate, with effects on local to international (transboundary) scales (Fowler et al., 2016). While substantial reductions in $SO_2$ emissions and limited reductions in $NO_x$ emissions have been achieved in Europe and North America following legislation designed to improve air quality, $NH_3$

emissions, primarily from the agricultural sectors (94 % of total $NH_3$ emissions in Europe in 2014) have seen much smaller reductions (EEA, 2016). In the period 2000-2014, $NH_3$ emissions are estimated to have decreased in the EU-28 (28 member states of the European Union) by only 8 % from 4.3 to 3.9 million tonnes, with the UK contributing 7.2 % in 2014 (EEA, 2016). $SO_2$ emission are estimated to have declined by 69 % and $NO_x$ by 39 % across the EU-28 over the same period.

$NH_3$ is known to contribute significantly to total nitrogen (N) deposition to the environment, and causes harmful effects through eutrophication and acidification of land and freshwaters. This can lead to a reduction in both soil and water quality, loss of biodiversity and ecosystem change (e.g. Pitcairn et al., 1998; Sheppard et al., 2011). In the atmosphere, $NH_3$ is the major base for neutralization of atmospheric acids, such as $SO_2$ and $NO_x$ emitted from combustion processes (vehicular and industrial), to form ammonium-containing particulate matter (PM): primarily ammonium sulphate ($(NH_4)_2SO_4$) and ammonium nitrate

($NH_4NO_3$). This secondary PM is mainly in the 'fine' mode with diameters of less than 2.5 µm (i.e. $PM_{2.5}$ fraction) (Vieno et al., 2014). The effects of PM on atmospheric visibility, radiative scattering (and the greenhouse effect) and on human health (bronchitis, asthma, coughing) are well documented (e.g. Kim et al., 2015; Brunekreef et al., 2015). Inputs of $NH_3$ and $NH_4^+$ (collectively termed $NH_x$) are the dominant drivers of ecological effects of deposited N, compared with wet deposited $NH_4^+$ in rain (UNECE, 2016) and the importance of $NH_x$ can be expected to increase further, relative to oxidised N, as $NO_x$ emissions

have been decreasing faster than $NH_3$ emissions (Reis et al., 2012; EEA, 2016; EU, 2016).

In gaseous form, $NH_3$ has a short atmospheric lifetime of about 24 hours (Wichink Kruit et al., 2012). It is primarily emitted at ground level in the rural environment, and is associated with large dry deposition velocities to vegetation (Sutton and Fowler, 2002). High $NH_3$ concentrations can lead to acute problems at a local scale to, for example, nature reserves located in intensive

agricultural landscapes (Sutton et al., 1998; Cape et al., 2009a; Hallsworth et al., 2010; Vogt et al., 2013). The $NH_3$ remaining in the atmosphere generally partitions to PM where the $NH_4^+$ can have a lifetime of several days (Vieno et al., 2014). Although $NH_4^+$ dry deposits at the surface, the primary removal mechanism for $NH_4^+$ is thought to be through scavenging of PM by cloud and rain, leading to wet deposition of $NH_4^+$ (Smith et al., 2000). Characterising the relationship between $NH_3$ emissions and the formation of PM is, however, not straight forward; an increase in $NH_3$ emissions does not automatically translate to a

proportionate increase in $NH_4^+$ (Bleeker et al., 2009). The relationship depends on climate and meteorology as well as the concentration of other precursors to PM formation such as $SO_2$ and $NO_x$ (Fowler et al., 2009). While it is clear that reductions in $NH_3$ emissions will lead to reductions in overall $NH_4^+$ concentrations, the relative changes in gaseous $NH_3$ and $NH_4^+$ particles remains poorly quantified.

International targets have been agreed to reduce $NH_3$ emissions to move towards protection against its harmful effects. These include the UNECE Convention on Long-Range Transboundary Air Pollution (CLRTAP) Gothenburg Protocol and the recently revised EU National Emission Ceilings Directive (NECD 2016/2284) (EU, 2016). The 1999 UNECE Gothenburg Protocol is a multi-pollutant protocol to reduce acidification, eutrophication and ground-level ozone by setting emissions ceilings for sulphur dioxide, nitrogen oxides, volatile organic compounds and ammonia, which are to be met by 2020. Revised

in 2012, the protocol requires national parties to jointly reduce emissions of $NH_3$, in the case of the EU-28 by 6 % between 2005 and 2020 (Reis et al., 2012). Under the revised NECD (EU, 2016), the EU is also committed to reductions of 6% for $NH_3$, but by a later date of 2029, as well as an additional 13% reduction in $NH_3$ emission beyond 2030 compared with a 2005 baseline.



Although this demonstrates that there is currently no strong commitment to reduce $NH_3$ emissions compared with $SO_2$ and $NO_x$, other supporting measures should also be noted including the Industrial Emissions Directive 2010/75/EU (IED), which requires pig and poultry farms (above stated size thresholds) to reduce emissions using Best Available Techniques. The IED

applies to around 70 % of the European poultry industry and around 25 % of the pigs industry (UNECE, 2010). In tandem, revised UNECE 'Critical Levels' (CLe) of $NH_3$ concentrations to protect sensitive vegetation and ecosystems were adopted in 2007 (UNECE, 2007). These set limits of $NH_3$ concentrations to 1 µg $NH_3$ m$^{-3}$ and 3 µg $NH_3$ m$^{-3}$ annual mean for the protection of lichens-bryophytes and other vegetation, respectively (Cape et al., 2009b). The new CLes replaced the previous single value of 8 µg $NH_3$ m$^{-3}$ (annual mean) and have since been adopted as part of the revised Gothenburg Protocol. Such CLes for $NH_3$

are widely exceeded, including over the areas designated as Special Areas of Conservation (SAC) under the Habitats Directive, and indicates a significant threat to the Natura 2000 network established by that directive (Bleeker et al., 2009; Hallsworth et al., 2010; van Zanten et al., 2017).

Few countries have established systematic networks to measure $NH_3$ across their domains. In the Netherlands, a continuous

wet annular denuder method (AMOR, replaced by the DOAS (Differential Optical Absorption Spectroscopy) device in 2015) has been used at 8 stations in the Dutch National Air Quality Monitoring Network (Van Pul et al., 2004; van Zanten et al., 2017). The Ammonia in Nature (MAN) network established in 2005 in the Netherlands monitors $NH_3$ with passive diffusion tubes in Natura 2000 areas (Lolkema et al., 2015). In the USA, the Ambient Ammonia Monitoring Network (AMoN) uses passive (Radiello) samplers at 50 sites since Oct 2010 (Puchalski et al., 2011). Hungary (Horvath et al., 2009), Belgium (den

Bril et al., 2011), Switzerland (Thöni et al., 2004), West Africa (Senegal and Mali under the Pollution of African Capitals program; Adon et al., 2016) and China (Xu et al., 2016) also have long-term $NH_3$ measurements (see review by Bleeker et al., 2009).

In the UK, the National Ammonia Monitoring Network (NAMN) was established in September 1996 with the aim of

establishing long-term continuous monthly measurements of atmospheric $NH_3$ gas (Sutton et al., 2001a). Particulate $NH_4^+$ measurements were added in 1999, since this was expected to exhibit different spatial patterns and temporal trends to gaseous $NH_3$ (Sutton et al., 2001b). The NAMN thus provides a unique and important long-term record for examining responses to changing agricultural practice and allows assessment of the compliance of $NH_3$ emissions with targets established by international policies on emissions abatement. Measurements of $NH_3$ and $NH_4^+$ in the NAMN also address spatial patterns,

covering both source and sink areas to test performance of atmospheric transport models, to support estimation of dry deposition of $NH_x$, to improve estimation of the UK $NH_x$ budget (Fowler et al., 1998; Smith et al., 2000; Sutton et al., 2001b) and to assist with the assessment of exceedance of critical loads and critical levels (UNECE, 2007).

This paper provides an analysis on the state of atmospheric concentrations of $NH_3$ and $NH_4^+$ in the UK from 1998 to 2014 and

their spatial and temporal trends. Overall, 17 years of continuous long-term $NH_3$ measurement data and 16 years of continuous long-term $NH_4^+$ measurement data from the NAMN are analysed to assess trends in concentrations in relation to estimated changes in emissions. The long-term measurement dataset is also used to explore spatial and temporal patterns in $NH_3$ and $NH_4^+$ across the UK in relation to regional variability in emission source sectors.





## 2 Material and Methods

### 2.1 Network structure and site requirements

The design strategy for NAMN was to sample at a large number of sites (>70) using low-frequency (monthly) sampling for cost-efficient assessment of temporal patterns and long-term trends. It was also recognised that the location of the network

sites needed to consider the extent of sub-grid variability and the representativeness of sampling points. Spatially detailed local-scale $NH_3$ monitoring was therefore also carried out at a sub-1 km level to assess the extent to which a monitoring location is representative (Tang et al., 2001b). The NAMN started with 70 sites. Over time, new sites were added to fill gaps in the map, some sites were closed following reviews and some sites had to be relocated due to local reasons, for example land ownership changes or site re-development. The number of sites peaked at 93 in 2000, but since 2009 has been stable at 85

sites. The locations of the NAMN sites for $NH_3$ and $NH_4^+$ in 2012 are shown in Figure 1a & b.

**<INSERT FIGURE 1>**

The selection of NAMN sites to provide a representative concentration field across the UK was aided by the availability of an

estimated UK $NH_3$ concentration field at a 5 km by 5 km grid resolution provided by the Fine Resolution Atmospheric Multi-pollutant Exchange (FRAME) model (Singles et al., 1998; Fournier et al., 2002). A comparison of FRAME modelled $NH_3$ concentrations for NAMN sites with FRAME modelled concentrations for the whole of the UK shows that the network has a good representation in the middle air concentration classes of 3-4 $\mu g\ m^{-3}$, but with an over-representation at high concentrations and under-representation at low concentrations (Figure 1c). Since air concentrations are more variable in high concentration

areas, a larger number of monitoring sites were located in these areas than in remote low concentration areas where air concentrations are more homogeneous. Similarly, the monitoring sites were strategically selected to cover source areas of expected high concentrations and variability on the basis of the FRAME model $NH_3$ concentration estimates (Figure 1a & b), and this approach was expected to provide additional evidence to test the performance of atmospheric dispersion models (Fournier et al., 2005; Dore et al., 2015). When compared with other atmospheric chemistry transport models, FRAME was

found to correlate well with measured $NH_3$ concentrations (Dore et al. 2015). The NAMN sites were also similarly checked for representativeness of particulate $NH_4^+$ by comparing FRAME modelled $NH_4^+$ concentrations at NAMN sites with modelled concentrations for the whole of the UK, which demonstrates a good representation across the range of expected concentrations (Figure 1d).

### 2.2 Atmospheric $NH_3$ and $NH_4^+$ measurements

Monthly time-integrated measurements of atmospheric $NH_3$ are made in the NAMN using a combination of passive samplers (Sutton et al., 2001a; Tang et al., 2001a) and an active diffusion denuder method referred to as the DEnuder for Long Term Atmospheric (DELTA) sampler (Sutton et al., 2001a &c). In terms of passive samplers, membrane diffusion tubes (3.5 cm long) with a limit of detection (LOD) around 1 $\mu g\ NH_3\ m^{-3}$ (Sutton et al., 2001a) were used in the first 4 years (September 1996 – April 2000). These were replaced in May 2000 with the more sensitive Adapted Low-cost, Passive High Absorption

(ALPHA, LOD = 0.03 $\mu g\ NH_3\ m^{-3}$) diffusive samplers (Tang et al., 2001a; Tang and Sutton, 2003), following a period of parallel testing (Sutton et al., 2001c).

Particulate $NH_4^+$ measurement was added to the NAMN in 1999 at all DELTA sites (50) in the first two years (1999 and 2000). Following this initial period, the sampling density was reduced during early 2001 to 37 sites and has been stable at 30 sites

since 2006. Although not presented in this paper, the DELTA samplers additionally provide concentrations of acid gases ($HNO_3$, $SO_2$, HCl) and aerosols ($NO_3^-$, $SO_4^{2-}$, $Cl^-$, $Na^+$, $Ca^{2+}$, $Mg^{2+}$) for the UK Acid Gas and Aerosol monitoring network





(AGANet) at a subset of NAMN DELTA sites (Tang et al., 2015; Conolly et al., 2016). Measurement data from the AGANet are used to aid interpretation of $NH_3$ and $NH_4^+$ results in Sect. 3.5.6.

### 2.2.1   DELTA method

The DELTA method uses a small pump to sample air (0.2 to 0.4 L min$^{-1}$) in combination with a high-sensitivity gas meter to record sampled volume (Sutton et al., 2001c). Two citric acid coated denuders (10 cm long borosilicate glass tubes) in series are used to collect $NH_3$ gas and to check the collection efficiency. A collection efficiency correction is applied to the measurement (Sutton et al., 2001d). The corrected air concentration ($\chi_a$ (corrected)) is determined as in Equation 1:

$$\chi_a \text{ (corrected)} = \chi_a \text{ (Denuder 1)} * \frac{1}{1-\chi_a\left[\frac{\chi_a \text{(Denuder 2)}}{\chi_a \text{(Denuder 1)}}\right]} \qquad (1)$$

Typically, denuder collection efficiency is better than 90% (Conolly et al., 2016). At 90 % collection efficiency, the correction represents < 1 % of the corrected air concentration. Individual measurements with collection efficiency < 75 % (correction amounts to 11 % of the total at 75%) are flagged as valid, but less certain (Tang and Sutton, 2003). Where less than 60 % of the total capture is recorded in the first denuder, the correction factor amounts to greater than 50 % and is not applied. The air concentration of ($\chi_a$) of $NH_3$ is then determined as the sum of $NH_3$ in denuders 1 and 2 (Equation 2):

$$\chi_a = \chi_a \text{ (Denuder 1)} + \chi_a \text{ (Denuder 2)} \qquad (2)$$

At sites where particulate $NH_4^+$ is also sampled, a 25 mm filter pack with a citric acid impregnated filter is added after the denuders to capture the $NH_4^+$. The calculated air concentrations ($Y_a$) of $NH_4^+$ is corrected for incomplete capture of $NH_3$ by

the double denuder. The corrected air concentrations ($Y_a$ (corrected)) of $NH_4^+$ is determined as in Equation 3:

$$Y_a \text{ (corrected } NH_4^+) = Y_a \text{ (}NH_4^+) - [((\chi_a \text{ (corrected } NH_3) - [(\chi_a \text{ (Denuder 1 } NH_3) + \chi_a \text{ (Denuder 2 } NH_3)])* (18/17)] \quad (3)$$

For $NH_4^+$ sampling, loss of $NH_3$ due to volatilisation of $NH_4^+$ from the acid impregnated filter has been investigated, by adding

a third citric acid coated denuder after the filter pack which was found to be negligible. At DELTA sites where additional simultaneous sampling of acid gases and particulate phase components are made for AGANet, ion balance checks between anions and cations in the particulate phase are performed to provide an indication of the quality of the particulate measurements. For the acid and base particulate components, close coupling is expected between $NH_4^+$ and the sum of $NO_3^-$ and $SO_4^{2-}$, as $NH_3$ is neutralised by $HNO_3$ and $H_2SO_4$ to form $NH_4NO_3$ and $(NH_4)_2SO_4$, respectively (Conolly et al., 2016).

At the Bush OTC site in Scotland (UK-AIR ID = UKA00128), duplicate DELTA measurements are made to assess the reproducibility of the method. For continuous monthly measurements between 1999 and 2014, the $R^2$ between the duplicate systems was 0.96 for both $NH_3$ and $NH_4^+$ (supp. Figure S1).

### 35   2.2.2   Passive methods

The $NH_3$ membrane diffusion tubes deployed in the NAMN from 1996 to 2000 are hollow cylindrical tubes (FEP, 3.5 cm long). A cap at the top end holds in place two stainless steel grids coated with sulphuric acid. The lower air-inlet end of the tube is capped with a gas-permeable membrane (Sutton et al., 2001a; Tang et al., 2001a; Thijsse, 1996). In comparison, the ALPHA passive sampler is a badge-type high sensitivity sampler with an uptake rate that is ~20 times faster than the diffusion




tube. It consists of a cylindrical low-density polyethylene body. An internal ridge supports a cellulose filter coated with citric acid, which is held in place with a polyethylene ring. The open end is capped with a PTFE membrane, providing a diffusion path length of 6 mm between the membrane and absorbent surface (Tang et al., 2001a).

5    Triplicate passive samplers are deployed for every measurement in the NAMN. Where the % coefficient of variation (CV) of the triplicate samplers is greater than 30% for the diffusion tubes or greater than 15% for the ALPHA samplers, the sample run is classed as failing the quality control test. Large discrepancies are most likely due to contamination of samples and data from contaminated samples are excluded from the assessment in this paper.

10    The passive methods are calibrated against the DELTA method in the NAMN by ongoing comparison at several sites representing a wide range of ambient $NH_3$ concentrations (see Sect. 2.2.4). Since 2009, the number of inter-comparison sites has been nine. These are Auchencorth (UKA00451), Bush OTC (UKA00128), Glensaugh (UKA00348), Lagganlia (UKA00290), Llynclys Common (UKA00270), Moorhouse (UKA00357), Rothamsted (UKA00275), Sourhope (UKA00347) and Stoke Ferry (UKA00317). The inter-comparison is used to establish a regression between the active and passive methods, with the DELTA samplers as the reference system, since the air volume sampled is accurately measured with high sensitivity gas meters. The calibration is necessary to account for the fact that the sampling path length in the passive samplers is longer than the distance between the membrane and adsorbent, due to the additional resistance to molecular diffusion imposed by the turbulence damping membrane at the inlet and the presence of a laminar boundary layer of air on the outside of the sampler (Tang et al., 2001a). In addition, parallel measurements were made at a high $NH_3$ concentration farm site (1998-2007) to extend the calibration range, and to ascertain linearity of response to high concentrations. To ensure that no bias is introduced in the sampling and to maintain the validity of long-term trends, the calibration is evaluated on an annual basis (Tang and Sutton, 2003; Conolly et al., 2016).

For the period up to 2000 when the diffusion tubes were implemented in the NAMN, their calibration (at 10 µg m$^{-3}$) amounts to an average of 1.5 % compared with the DELTA system. The mean ALPHA sampler calibration (at 10 µg m$^{-3}$), compared with the DELTA system, amounts to a correction of 10 % (ALP1: prototype 1, 1998-2000), 15 % (ALP2: injection mould 1, 2001-2005), 17 % (ALP3: injection mould 2, 2006), 34 % (ALP4: injection mould 2 + new membrane, 2007-2008) and 40 % (ALP5: injection mould 2 + new membrane + new lab/instrument FloRRia, 2010-2014), respectively. The new PTFE membrane (5 µm pore size) is supported on a regular polypropylene grid and is thicker (305 µm) than the earlier PTFE membrane (also 5 µm pore size, but 265 µm thickness) used which was supported instead on a randomly arranged polypropylene support material. The difference in calibration was therefore due to the extra resistance to gas diffusion imposed by the new thicker membrane. The annual calibration of the methods shows both high precision and constancy between years (Figure 2), which is important to support the detection of temporal trends in $NH_3$ concentrations. There is no systematic trend over time in either of the passive method calibrations.

**<INSERT FIGURE 2>**

The comparison of monthly measurement data between the DELTA and calibrated passive measurements demonstrated a close agreement (Figure 3). The correlation ($R^2$) between DELTA and calibrated diffusion tubes was 0.91 (Figure 3a), while the correlation between DELTA and calibrated ALPHA samplers was 0.92 (Figure 3b). From the calibrated results, the intercept for the diffusion tubes was 0.10 µg $NH_3$ m$^{-3}$, while that for the ALPHA samplers was 0.03 µg $NH_3$ m$^{-3}$, demonstrating the improvement in sensitivity with the ALPHA samplers compared with the diffusion tubes (Figure 3). In the present case the value of the intercepts, even for diffusion tubes, is much less than typical $NH_3$ air concentrations (see Sect. 3). However, this cannot be assumed to be the case in other implementations of the same methods. Experience from other studies using the lower





sensitivity diffusion tubes indicates a tendency to overestimate $NH_3$ concentrations under clean conditions (RGAR, 1990; Thijsse et al., 1996; Tang et al., 2001a; Lolkema et al., 2015). This observation points to the need for any application of $NH_3$ passive sampling for ambient monitoring to be accompanied by testing and calibration against a verified active sampling method. In independent assessments, for example in the USA (Puchalski et al., 2011), the ALPHA samplers performed well

against a reference annular denuder method with a median relative percent difference of $-2.4\%$.

**\<INSERT FIGURE 3\>**

### 2.2.3    Chemical analysis

$NH_3$ gas captured on the acid coating of the denuder (DELTA), grid (diffusion tubes) or filter paper (ALPHA), and particulate $NH_4^+$ captured on the DELTA aerosol filter, are extracted into deionised water and analysed for $NH_4^+$ on an ammonia flow injection analysis system. The analytical instrument has changed over the network's operational period from the AMFIA (ECN, NL) to the FloRRIA (Mechatronics, NL), an updated model based on AMFIA (Conolly et al., 2016). The principles of operation of both instruments are the same and are based on selective diffusion of $NH_4^+$ across a PTFE membrane at $c$. pH 13

into a counter-flow of deionized water, allowing selective detection of $NH_4^+$ by conductivity (Wyers et al., 1993).

### 2.2.4    Data Quality Control

Measurement data are checked and screened, based on the quality management system applied in the UK air monitoring networks (Tang and Sutton, 2003). Data quality is assessed against the following set quality control criteria:  a) DELTA system: monitoring of the air flow rate and the use of two denuders in every sample to assess capture efficiency for $NH_3$, and b) passive

samplers: use of triplicate samplers for monitoring $NH_3$ concentrations at every site, to allow an assessment of sampling precision, and c) ongoing calibration of passive samplers against the DELTA. Data flags are applied to the dataset; a full list of these is available from the EMEP website (http://www.nilu.no/projects/ccc/flags/index.html). Following the quality control checks and data flagging on the collected dataset, the annually ratified data from the NAMN are made publically available on the Department for Environment, Food & Rural Affairs (Defra) UK-AIR website (https://uk-air.defra.gov.uk/) and are also in

the process of being made available on the EMEP website (http://ebas.nilu.no/).

### 2.2.5    Trend Analyses

Trend analyses were carried using (i) linear regression (LR), (ii) Mann-Kendall (MK) test (Gilbert, 1987) on annually averaged and monthly mean data, and (iii) Seasonal Mann-Kendall (SMK) test (Hirsch et al., 1982) on monthly data only. Mann-Kendall

tests were performed using the 'Kendall' package (McLeod, 2015) in the R software. Computation of the Sen's slope and confidence interval (for non-seasonal Sen's slope only) of the linear trend were performed using the R 'Trend' package (Pohlert, 2016). The SMK test (Hirsch et al., 1982) takes into account a 12 month seasonality in the time series data by computing the MK test on each of monthly 'seasons' separately, and then combining the results. So for monthly 'seasons', January data are compared only with January, February only with February, etc. No comparisons are made across season

boundaries.

The Sen's slope is the fitted median slope of a linear regression joining all pairs of observations. For the SMK, an estimate of the seasonal Sen's trend slope over time is computed as the median of all slopes between data pairs within the same season (i.e. January compared only with January etc.). Therefore no cross-season slopes contribute to the overall estimate of the SMK

trend slope. The main advantages of the MK approach over linear regression for trend assessments are that (i) it does not





require normally distributed data, (ii) it is not affected by outliers, and (iii) it removes the effect of temporal auto-correlation in the data. The MK approach are widely used in environmental time series assessments, e.g. long-term trends in precipitation (Serrano et al. 1999) and long-term trends in European air quality (EMEP, 2016; Torseth et al., 2012).

**3    Results and discussion**

In order to summarise and discuss the NAMN dataset, the spatial patterns in the measurements of $NH_3$ and $NH_4^+$ are considered in Sect. 3.1 (comparison with emission estimates) and Sect. 3.2 (comparison with modelled concentration estimates), seasonal patterns are discussed in Sect. 3.3, and long-term trends across the UK in Sect. 3.4.

**3.1    Spatial variability in $NH_3$ and $NH_4^+$ concentrations in relation to estimated emissions**

As a primary pollutant emitted from ground-level sources, $NH_3$ exhibits high spatial variability in concentrations (Sutton et al., 2001b; Hellsten et al., 2008; Vogt et al., 2013), confirmed by $NH_3$ data from the NAMN (e.g. range of $0.06 - 8.8$ µg m$^{-3}$ annual mean in 2005) (Figure 4a). The observed variability is consistent with the large regional variability in $NH_3$ emissions and sources (Figure 4c & d). With agriculture being the main source of $NH_3$ emissions, Figure 4a shows the largest concentrations of measured $NH_3$ in parts of the UK with the highest livestock emissions, such as eastern England (East Anglia),

north-west England (Eden Valley, Cumbria) and the border area between England and Wales (Shropshire) (Figure 4d). By contrast, the lowest $NH_3$ measured concentrations are found in the north-west Scottish Highlands (< 0.2 µg m$^{-3}$), which is consistent with the emissions map (Figure 4c). The 2005 data show exceedance of the Critical Levels for annual mean $NH_3$ concentrations of 1 and 3 µg $NH_3$ m$^{-3}$ for the protection of lichens-bryophytes and vegetation, respectively (UNECE, 2007) at many of the sites (53 % > 1 µg $NH_3$ m$^{-3}$ and 13 % > 3 µg $NH_3$ m$^{-3}$). In 2014, exceedance of the 1 and 3 µg $NH_3$ m$^{-3}$ CLe

increased to 60 % and 16 %, respectively. The widespread exceedance of the CLe for $NH_3$ concentrations across the UK thus represents an ongoing threat to the integrity of sites designated under the Habitats Directive, as well as nationally designated Sites of Special Scientific Interest (SSSI) and other sensitive habitats.

Concentrations of $NH_4^+$ are less spatially heterogeneous than those of $NH_3$, based on data from 30 sites (e.g. range of 0.14 to

1.8 µg m$^{-3}$ annual mean in 2005) with a more coherent pattern of variation across the country, reflecting regional differences in $NH_3$ concentrations (Figure 4b). Thus there is a general decreasing gradient from the south-east to the north-west of the UK, due to both $NH_3$ sources in England and import of particulate matter from Europe (Vieno et al., 2014; Dore et al., 2015). The limited variation across the UK for the annual average $NH_4^+$ concentrations can be attributed to the atmospheric formation process (providing a diffuse source) and its longer atmospheric lifetime.

**<INSERT FIGURE 4>**

A similar picture is reported by the Dutch National Air Quality Monitoring Network (van Zanten et al., 2017), with large spatial variability of $NH_3$ concentrations ($2 - 20$ µg $NH_3$ m$^{-3}$) across the country and a more homogeneous distribution of

particulate $NH_4^+$ (1-2 µg $NH_4^+$ m$^{-3}$ in 2014), although the number of Dutch monitoring sites reported there is much smaller with only 8 stations providing continuous measurements. Both $NH_3$ and $NH_4^+$ concentrations were correlated with emission density, but the correlation was smaller for $NH_4^+$ than for $NH_3$ because of the larger contribution to $NH_4^+$ concentrations from long-range transport in the Netherlands.



The UK $NH_3$ emissions inventory is calculated and spatially distributed annually. Agricultural sources at a 5 km by 5 km grid resolution are combined with a large number of non-agricultural sources (Sutton et al., 2000; Tsagatakis et al., 2016) at a 1 or 5 km resolution to produce the annual $NH_3$ emissions data, and maps at a 1 km by 1 km grid resolution are reported by the official UK National Atmospheric Emissions Inventory (NAEI; http://naei.defra.gov.uk/data/mapping). In the UK, agriculture

accounts for > 80% of total $NH_3$ emissions and is estimated by the National Ammonia Reduction Strategy Evaluation System (NARSES) model (Webb & Misselbrook 2004; Misselbrook et al., 2015). For the agricultural $NH_3$ emission maps, parish statistics on livestock numbers and crop areas are combined with satellite-based land cover data to model emissions at a 1 km resolution, using the AENEID model (Dragosits et al., 1998; Hellsten et al., 2007). For reasons of data confidentiality, the 1 km data need to be aggregated to produce annual agricultural $NH_3$ emissions maps at a 5 km by 5 km grid resolution. National

emission estimates for $NH_3$ are submitted to both the European Commission under the NECD (2001/81/EC) and the United Nations Economic Commission for Europe (UN/ECE) under the Convention on Long-Range Transboundary Air Pollution (CLRTAP).

The AENEID approach (Dragosits et al. 1998) can further be used to classify each 5 km by 5 km grid square in the UK into

dominant $NH_3$ emission source categories (Figure 4d), following the method of Hellsten et al. (2008), where grid squares with >45% from a given category are referred to as dominated by that source. The seven categories are: cattle, pigs & poultry (combined for data disclosivity reasons), sheep, fertilizer application to crops and grassland, non-agricultural sources, as well as a mixed category where no single source dominates, and background. Background grid squares are defined by very low $NH_3$ emissions of <1 kg N ha$^{-1}$ y$^{-1}$.

Using the dominant emission sources map, each site in the NAMN is classified to one of the seven categories just described. This provides information of the main emission source type expected in the 5 km by 5 km grid square containing the monitoring site and is useful for assessing whether the network has a good representation of key emission source categories (Supp. Figure S2a & b). Over the period since the NAMN was established, from 1996 to present, there have been substantial changes in

emissions estimated for the different source sectors. For analysis in this paper, the dominant sources map for 2005 emission year was used as representing the mid-point of the data series (1998-2014) and compared with the classification from other years for consistency. This categorization of sites is used further in the interpretation of the monitored $NH_3$ and $NH_4^+$ concentrations and their long-term trends in the next sections.

**<INSERT FIGURE 5>**

### 3.2 Spatial variability in $NH_3$ and $NH_4^+$ concentrations in relation to modelled concentrations

Comparison of measurements with modelled $NH_3$ concentrations from the FRAME model for an example year of 2012 showed significant scatter when considering the full network of sites ($n = 85$, $R^2 = 0.62$) (Figure 5a). In this graph, each point is colour-

coded according to the estimated dominant $NH_3$ emission source category for the 5 km by 5 km grid square. This updates a similar comparison from Sutton et al. (2001b) for the year 2000. The scatter may be explained by the large local spatial variability of $NH_3$, related primarily to rapid decreases of $NH_3$ concentrations with distance from a source (see e.g. Pitcairn et al., 1998; Dragosits et al., 2002), with the result that a single site measurement only gives an approximate indication of concentrations across the model grid square it is located in. At many of the sites where the model overestimates concentrations,

the measurements are in fact carried out in nature reserves, or in clearings inside forests. The monitoring sites in these sink areas are typically well away from local sources. Conversely, some of the outliers where measurements are larger than the



model predictions show indications of being affected by nearby emission sources, as was established by investigations during site visits.

**<INSERT FIGURE 6>**

Figure 6 considers measured $NH_3$ concentrations at a subset of sites (44 out of the full 85 sites) that are located away from nearby local sources, in forest or semi-natural areas, following the site classification and assessment by Hallsworth et al. (2010). For this restricted set of sites, $R^2 = 0.76$ for 2012 which is higher than the correlation for the overall UK network. The improvement in correlation between measured and modelled $NH_3$ concentrations for this subset of sites can be explained by

the monitoring locations typically being further away from sources, so that uncertainties in local emissions estimates are to some extent averaged out. This observation is also consistent with the findings of Vieno et al. (2009).

In contrast to $NH_3$, the correlation between NAMN measurements and FRAME model output is stronger for particulate $NH_4^+$ concentrations ($R^2 = 0.87$). However, measured concentrations are generally larger than the modelled ones (slope 1.1, intercept

$-0.16$ µg m$^{-3}$ (Figure 5b). One reason for the better agreement for $NH_4^+$ is the more slowly changing spatial patterns in concentrations, which are not expected to vary on a finer scale than the model's 5 km by km grid, improving the representativeness of site-based measurements. The 2012 comparison shown here updates an earlier inter-comparison assessment carried out by Dore et al. (2007) for the year 2002 and demonstrates that the FRAME model is performing well in describing the spatial distribution of $NH_4^+$. However, for the 2012 inter-comparison, the FRAME model appears to

underestimate $NH_4^+$ at sites with concentrations $< 0.6$ µg $NH_4^+$ m$^{-3}$, with better agreement at concentrations above 0.6 µg $NH_4^+$ m$^{-3}$. This suggests either too low a formation rate for $NH_4^+$ in the model at cleaner sites, or too high a removal rate for $NH_4^+$, or a combination of both. The presence of higher measured $NH_4^+$ concentrations in remote areas than shown by the model may also indicate that $NH_4^+$ has a longer residence time than treated in the model. Similar regressions between NAMN and FRAME $NH_4^+$ aerosol concentrations were observed for other years. For example, for 2008 the FRAME model underestimated $NH_4^+$

at concentrations $< 0.7$ µg $NH_4^+$ m$^{-3}$ (slope 1.2, intercept $-0.26$ µg$^{-3}$; $R^2 = 0.89$, range $= 0.2 - 1.4$ µg m$^{-3}$). Changes in the chemical climate, such as reduced emissions of $SO_2$ in the UK, are postulated to affect conversion rates of $NH_3$ into $NH_4^+$, as well as the dry deposition rates, leading to more $NH_3$ remaining in the atmosphere (van Zanten et al., 2017). This is discussed further in Sect. 3.5.6.

### 3.3 Seasonal variability in measured UK $NH_3$ and $NH_4^+$ concentrations

A comprehensive account of the seasonal variability of $NH_3$ and $NH_4^+$ for different regions across the UK is provided by the NAMN. In Figure 7, the average seasonal cycles of grouped sites from four different emission source categories are compared for $NH_3$ and $NH_4^+$.

**<INSERT FIGURE 7>**

In addition to substantial differences in the overall magnitude of $NH_3$ concentrations, where the largest concentrations in the network are found at sites dominated by pig and poultry farming, followed by areas where cattle farming predominates, it is clear that the seasonal patterns of $NH_3$ also vary depending on the dominant source type (Figure 7a). For background sites (defined as located in grid squares with $NH_3$ emissions $<1$ kg N ha$^{-1}$ y$^{-1}$), a clear summer maximum in $NH_3$ concentrations can be observed, with minimum concentrations occurring in winter. The summer peak is probably related to increased land surface

$NH_3$ emissions in warm, dry summer conditions, both from the presence of low-density grazing livestock and wildlife. It is also related to surface factors such as the compensation point for vegetation, which is defined as the concentration below which





growing plants start to emit $NH_3$ into the atmosphere (Sutton et al., 1995). The interaction between atmospheric $NH_3$ concentrations and vegetation is complex, leading to both emission and deposition fluxes, depending on relative differences in concentrations. However, it is well established that warm, dry conditions promote $NH_3$ emission from vegetation (e.g. Massad et al., 2010; Flechard et al., 2013). It is therefore possible that bi-directional exchange with vegetation is at least partly

controlling $NH_3$ concentrations at remote sites distant from intensive livestock farming.

The possibility for such interactions can be considered further using the example of Inverpolly (UKA00457), a remote background site in the NW Scottish Highlands. This site shows a very clear seasonal cycle with peak concentrations in July when warmer, drier conditions prevail, while lowest concentrations occur during the cooler and wetter winter months (Figure

8a & b). A smaller peak in $NH_3$ can also be seen annually in April, which indicates potential longer range influences of manure spreading in spring, even at this remote location (Figure 8b). Although there is substantial scatter, Figure 9 shows that there is significant correlation between monthly $NH_3$ concentrations and both temperature ($R^2 = 0.33$, $n = 231$, $p < 0.05$) and precipitation ($R^2 = 0.19$, $n = 231$, $p < 0.05$). The influence of temperature and rainfall on $NH_3$ emission and concentrations is well characterised (e.g. see Sutton et al., 2013; van Zanten et al., 2017).

**\<INSERT FIGURE 8\>**

**\<INSERT FIGURE 9\>**

For sites dominated by emissions from sheep farming, the seasonal profile in $NH_3$ concentrations is similar to that for background sites, although the summer maximum in $NH_3$ is larger than background sites, because grazing emissions are larger (Hellsten et al., 2008). It is notable that the peak $NH_3$ concentration occurs later in the year for background areas (July-September) than for sheep areas (June-August). This may be related to the seasonal presence of lambs, which are often only present for the first part of the summer. In areas with more intensive livestock farming, where emissions comes from either

cattle or from pig & poultry farming, the largest concentrations are observed in spring and autumn, corresponding to periods of manure application to land. The spring peak in March is larger than the autumn peak in September, which coincides with the main period for manure application being in spring, before the sowing of arable crops or early on in the grass-growing period (Hellsten et al., 2007). Ammonia concentrations in these areas are also larger in summer than winter, due to warmer conditions promoting volatilization. Interestingly, the dip in concentrations in June matches a period when crops will be

actively growing with possible uptake and removal of $NH_3$ from the atmosphere.

For particulate $NH_4^+$, as expected for a secondary pollutant, concentrations are more decoupled from the dominant $NH_3$ source sectors in the vicinity of a site. The seasonal trends in $NH_4^+$ are broadly similar for the four source sectors shown in Figure 7b, with the magnitude of the $NH_4^+$ concentrations reflecting $NH_3$ concentrations at a regional level. Here the spring peak is more

driven by long-range transboundary transport, e.g. influence from continental Europe (Vieno et al., 2014). Nevertheless, it is notable that the winter minima for $NH_4^+$ aerosol concentrations at sheep and background sites are more pronounced than that for pig, poultry and cattle dominated sites. This may be a result of a combination of smaller $NH_3$ emissions in winter in these areas (as indicated by Figure 7a) and differences in long-range transport to the more remote areas in winter conditions.

Overall, the seasonal distributions show that $NH_3$ concentrations are mostly governed by local emission sources and by changes in environmental conditions, with warm, dry weather favouring increased volatilisation. By contrast, particulate $NH_4^+$ concentrations are largely determined by more distant sources through long-range transport and synoptic meteorology.





### 3.4 Long-term trends in estimated UK NH₃ emissions

UK NH$_3$ emissions are estimated to have fallen by 16 % between 1998 and 2014, from 336 to 281 kt (Figure 10a) (http://naei.defra.gov.uk/). The most significant cause of the estimated reductions has been decreasing cattle, pig and poultry numbers in the UK over this period. Between 2013 and 2014, the decreasing trend in UK NH$_3$ emissions was however reversed

with an increase of 3.3 % from 272 to 281 kt NH$_3$ due to an increase in emissions from the agricultural sector from 224 kt in 2013 to 234 kt in 2014. This is attributed to an increase in dairy cow numbers (and dairy cow N excretion) and increase in fertiliser N use (particularly urea, which is associated with a higher emission factor than other fertilisers types used in the UK) (Misselbrook et al. 2015; http://naei.defra.gov.uk/).

**\<INSERT FIGURE 10\>**

Although the UK met the 2010 emission ceilings target of 297 kt NH$_3$ emission per year set out under the Gothenburg Protocol and NEC Directive, it is committed to a further emission reduction by 2020 of 8 % from the 2005 total under the 2012 revised Gothenburg Protocol, and by 17% after 2030 under the revised 2016 NEC Directive (EU, 2016). The revised 2020 target of

282 kt NH$_3$ (8% reduction of the baseline figure of 307 kt NH$_3$ emissions total in 2005) may require emission strategies to be implemented, rather than relying on decreasing livestock populations as during the recent decades.

Agricultural emissions are by far the largest NH$_3$ sources in the UK's emission inventory, accounting for 86 % and 83 % of the total NH$_3$ emissions in 1998 and 2014, respectively. The primary source of agricultural emissions is livestock manure

management, in particular from cattle which make up approximately 46 % of the total agricultural emissions, followed by pigs and  poultry contributing another 18 % in 2014 (Defra, 2015; Misselbrook et al., 2015) (Figure 10b). Over the period 1998 to 2014, NH$_3$ emissions from cattle are estimated to have decreased by 11 % (from 144 to 128 kt), with emissions estimated to have remained relatively stable since 2008, followed by a modest 2 % increase between 2013 and 2014 from 125 kt to 128 kt (Figure 10a; Figure 16). Emissions from pigs and poultry showed a large downward trend between 1998 and 2014, with a

decrease of 39 % (from 82.7 kt to 50.3kt) (Figure 10a; Figure 16), although the decreasing trend was reversed between 2012 and 2014, with an increase of 6 % from 46.7 kt to 50.3 kt, The sheep sector is a minor source, contributing 3.6 % to the total agricultural emissions. NH$_3$ emissions from this sector are estimated to have decreased by 24 % in 2014 relative to 1998 (from 13.3 to 10.1 kt).

### 3.5 Long-term trends in measured NH₃ concentrations

The UK NAMN dataset was analysed to compare levels and trends against the NH$_3$ emission inventory. To avoid bias due to changes in the number and locations of sites over the duration of the network, sites with incomplete data runs over selected periods for analysis are excluded. Based on these exclusion criteria, the number of sites with complete data runs was 59 for the period 1998 to 2014, 66 sites for 1999 to 2014, and 75 sites for the period 2000 to 2014.  To ensure consistency in the trend

analysis, several combinations of the available data were used:

1a. 1998 – 2014 (59 sites): annually averaged data

1b. 1998 – 2014 (59 sites): monthly mean data

2a. 1999 - 2014 (66 sites): annually averaged data

2b. 1999 - 2014 (66 sites): monthly mean data

3a. 2000 - 2014 (75 sites): annually averaged data

3b. 2000 – 2014 (75 sites): monthly mean data





A visualization of the time series according to dataset 1a is summarized in Figure 11. This shows the mean UK monitored annual $NH_3$ concentrations of 59 sites with complete data runs from 1998 (first complete year of monitoring) to 2014, summarised in a boxplot, together with annual mean UK rainfall and temperature data and compared with $NH_3$ emissions trends over the same period. The interquartile ranges and the spread of the $NH_3$ concentrations can be seen to be variable from year to year, demonstrating both substantial inter- and intra-annual variability.

**<INSERT FIGURE 11>**

### 3.5.1 Mann-Kendall non-parametric time series analysis

To detect trends and to indicate the significance level of the trends in the long-term NAMN data, the non-parametric Mann-Kendall (MK) approach was used combined with the Sen's slope method for estimating the trend and confidence interval of the linear trend (see Sect. 2.2.5). The classic MK test was used on the annually averaged data (datasets 1b, 2b, 3b), while both the classic MK and seasonal Mann-Kendall (SMK) tests were applied to the monthly averaged data (datasets 1a, 2a, 3a).

Results of the Mann-Kendall tests are summarised in Table 1. For each time series, the median annual trend (in units of $\mu g$ $NH_3^{-1} y^{-1}$) is estimated from the Sen's slope and intercept of the MK linear trend. To assess the relative change over time, the % relative median change was calculated from the estimated $NH_3$ concentration at the start ($y_0$) and at the end ($y_i$) of the selected time period ($100*[(y_i-y_0)/y_0]$) computed from the Sen's slope and intercept. This approach was adopted instead of a direct comparison of actual observed $NH_3$ concentrations at the start ($y_0$) and at the end ($y_i$) of the time series, since there is substantial inter-annual variability in the data (Figure 10, Figure 16). Using the estimated concentrations at the start and end from the fitted Sen's slope allows using a reference that is less sensitive to inter-annual variability than the actual observed concentrations.

**<INSERT TABLE 1>**

For the annually averaged $NH_3$ concentrations across the UK, dataset 1a (1998-2014, 59 sites) show a small, but non-significant decreasing trend (relative median change = −6.3 %), while datasets 2a (1999-2014, 66 sites) and 3a (2000-2014, 75 sites) show no discernible trends (median relative change = 0.0 % for both) (Table 1). Results from the analysis of monthly data from all three different data groupings (1b, 2b, 3b) (relative median change = −4.2 to −8.2 %) are similar to results for dataset 1a, based on analysis of annual data (Table 1). In the SMK tests on monthly data, two monthly "seasons" (January and April) in dataset 1b (1998-2014, 59 sites) are significant ($p < 0.05$) with a third monthly "season" (August) near-significant at $p = 0.06$. For datasets 2b (1999-2014, 66 sites) and 3b (2000-2014, 75 sites), August is the only monthly "season" in either time series to be close to significance at $p = 0.06$. Trends in individual monthly "seasons" are therefore weak and results between the MK and seasonal MK tests on monthly data are similar (Table 1).

### 3.5.2 Linear regression parametric time series analysis

**<INSERT TABLE 2>**





The parametric linear regression time series trend analysis was also performed on the different data groupings. Results of the linear regression tests are summarised in Table 2, and a comparison of trends from the Mann-Kendall with the linear regression approach is provided in Figure 12 for annual datasets 1a, 2a, 3a, and Figure 13 for monthly datasets 1b, 2b, 3b. A similar approach to the Mann-Kendall was taken to assess the relative change, by calculating the % relative change from the estimated

$NH_3$ concentration at the start ($y_0$) and at the end ($y_i$) of the time series ($100*[(y_i-y_0)/y_0]$) computed from the linear regression slope and intercept. The different data groupings all show small, but non-significant decreasing trends (relative change = $-2.4$ % to $-5.3$ %), similar to the trends and % relative median change from the MK and SMK analysis (Figure 12, Figure 13). This suggests that the the errors in the NAMN data are normally distributed and that no or few outliers are present, since the results from the non-parametric Mann-Kendall are very similar to the parametric least squares linear regression.

**<INSERT FIGURE 12>**

**<INSERT FIGURE 13>**

### 3.5.3    Trends in $NH_3$ concentrations *vs* trends in $NH_3$ emissions

Overall, the long-term $NH_3$ concentration data from the UK NAMN suggests evidence of a small, but non-significant decreasing trend (Figure 12, Figure 13). The level of reduction observed in the datasets is however less than the 16.3 %, 15.6 % and 13.1 % reduction in estimated UK $NH_3$ emissions over the periods 1998-2014, 1999-2014 and 2000-2014, respectively (Tables 1,2). Inventories have inherent uncertainties such as uncertainties in activity data and emission factors, or may be missing emission sources. In terms of measurement data, it has already been shown in Sects. 3.1 and 3.3 that the annually

averaged data mask considerable spatial and seasonal variability in $NH_3$ concentrations. Drivers contributing to this variability include the influence of climate on emissions, variations in management practice for a particular emission source, and influence of local emission sources and interactions on concentrations at a site. In addition, once emissions have taken place, the resulting atmospheric $NH_3$ concentrations are influenced by local deposition, which is in turn affected by receptor surfaces and by concentrations of interacting chemical species that affect atmospheric lifetime and transport distance of $NH_3$ and physical

dispersion (e.g. Bleeker et al., 2009; Sutton et al., 2013). In the following sections, we consider the possibility of interactions with climate, emission source type and chemical interactions as this may affect long term trends in $NH_3$ concentrations.

### 3.5.4    Influence of climate

UK temperature and rainfall varied from year to year over the period 1998 to 2014 (Figure 11), with no clear relationship with

$NH_3$ easily visible in the graph. Plotting the annual mean $NH_3$ concentrations against the average temperature and rainfall however does show indicatively that elevated annual mean $NH_3$ concentrations are observed in warmer years, and reduced annual mean $NH_3$ concentrations are observed in wetter years (Supp. Figure S3). This analysis for the full network is therefore consistent with the observation at a remote site (Inverpolly, Figure 9). The thermodynamic equilibrium shifts $NH_3$ from the aqueous (or particulate) phase to the gas phase with increased temperature, hence emissions from animal manures, soils and

vegetation increase with increasing temperature (Asman et al., 1998; Sutton et al., 1993). Conversely, increases in precipitation decrease $NH_3$ emissions because rain events dilute the available $NH_3$ pool, while having the potential to wash urea and $NH_x$ in solution from the surface. As $NH_3$ is soluble and washed out of the atmosphere by rainfall, this should also contribute to reduced $NH_3$ concentrations during wet periods.





An exception to this relationship can occur where N is excreted as uric acid from birds (e.g. poultry). In this case, sufficient water is needed to allow hydrolysis to form $NH_3$ (Riddick et al., 2014). In this situation, the arrival of rain promoted uric acid hydrolysis from seabird guano surfaces, which was limited in the absence of soil moisture. It is possible that this interaction could lead to $NH_3$ emissions from field spreading of poultry litter to be larger in wetter years. In a recent trend analysis of $NH_3$

concentrations from the Dutch Air Quality Monitoring Network, an attempt was also made to correct for meteorological (temperature and rainfall) influences for the eight monitoring stations, which broadly produced similar results with slightly enhanced statistical significance for the trends (van Zanten et al., 2017).

### 3.5.5   Influence of local emission sources

**<INSERT FIGURE 14>**
**<INSERT FIGURE 15>**

The inter- and intra-annual variability is also expected to be linked to influences from local emission source and activities. It has already been shown in Sect. 3.1 that the concentrations of $NH_3$ in air are greatest in parts of the country with a large

presence of livestock farming, particularly in areas of pig, poultry and cattle farming. Using the classification of NAMN sites according to dominant emission source sectors described in Sect. 3.1, the long-term change in $NH_3$ concentrations at sites grouped into four different emission source sectors: background, sheep, cattle, and pigs and poultry are compared in Figure 14 (annual mean data) and Figure 15 (monthly mean data). Results of the Mann-Kendall time series trend analysis are summarised in Table 3 and results of linear regression analysis are summarised in Table 4. A comparison of trends in measured $NH_3$

concentrations with trends in $NH_3$ emissions for the different source types then provided indicative evidence to support and inform the national emission inventory compilation.

**<INSERT TABLE 3>**
**<INSERT TABLE 4>**

For the 17 sites in cattle dominated areas, there is an increasing, but non-significant trend. Overall, based on MK analysis of annual data, the relative change from 1998 to 2014 is a 12 % increase (Table 3, Figure 14), compared with a smaller increase of 4 % from linear regression (Table 4, Figure 14). With the monthly data, there is no discernible trend (–0.9 % (MK); 1.4 % (LR)). In the seasonal MK test on monthly data (% relative median change = 3.9 %), no monthly "seasons" are significant,

with only January approaching significance at $p = 0.07$. The near-significant trend for January is likely to be due to unusually high $NH_3$ concentrations recorded in January at some sites in the first few months of the time series, attributed to manure spreading activities taking place in the winter months when the ground was frozen (confirmed by local observations), in direct contravention of good farming practice. In terms of UK cattle $NH_3$ emission, this has a decreasing trend with an estimated 11% decrease since 1998 (Figure 16, Table 5), and is therefore clearly in contrast to the non-discernible or small increasing

trend (non-significant) in $NH_3$ concentrations from cattle sites. In principle, a signal related to substantial livestock changes associated with the 2000 outbreak of Foot and Mouth Disease might have been expected. However, this outbreak was actually rather localized in north-west England and south-west England, and was followed by substantial restocking from 2001 (Sutton et al., 2006) and there was no detectable signal of FMD in the average for cattle-dominated areas.

**<INSERT FIGURE 16>**
**<INSERT TABLE 5>**



By contrast, in pig and poultry dominated areas (9 sites) there is a decreasing trend with significant reduction in measured $NH_3$ concentrations between 1998 and 2014 (−22 % (MK), $p = 0.02$, Table 3; −21 % (LR), $p = 0.06$, Table 4) from analysis of annual data (Figure 14). For the monthly data, the overall change based on linear regression is also a 22 % decrease ($p = 0.02$)

(Table 4, Figure 15), compared with a larger level of decrease based on MK analysis (−32 %, $p = 0.01$) (Table 3, Figure 15). The SMK test also show a significant decreasing trend (−11 %, overall $p < 0.001$), with 6 of the 12 monthly "seasons" showing significant trends (Feb, Jun, Nov, Dec: p <0.05, Oct: p < 0.01, Jan: p < 0.001). A decrease in emissions from pig and poultry of 39 % between 1998 and 2014 (Figure 16, Table 5) is therefore broadly supported, although not matched by a similar decrease in measured $NH_3$ concentrations.

For sheep dominated sites (4 sites), there is an increasing trend in $NH_3$ (MK: +16 %, $p = 0.17$, Table 3; LR: 20 %, $p = 0.09$, Table 4) between 1998 and 2014 in the annual data (Figure 14). The monthly data also show a similar upward trend (Figure 14) with relative change in concentrations of +19% based on MK ($p = 0.10$) (Table 3) and +17% based on LR ($p = 0.14$) (Table 4). The increasing trend at sheep sites is therefore in contrast to the estimated 24 % decrease in $NH_3$ emissions from this sector

since 1998 (Figure 16, Table 5). For the SMK test, no individual monthly "seasons" were significant, although 3 of the monthly "seasons" approached the significance level (Apr, Dec: $p = 0.08$, Oct: $p = 0.09$). Overall, the increasing trend from the SMK test is significant at $p < 0.01$. While the Sen's trend slope from both MK and SMK tests were comparable, at 0.0036 and 0.0033 µg $NH_3$ y$^{-1}$, respectively, the % relative median change results computed from them are very different (MK = 16 % $cf$ SMK = 210 %), because the intercepts of the fitted Sen's trend slopes are different (MK = 0.289 µg $NH_3$ m$^{-3}$ $cf$ SMK = −0.0267 µg

$NH_3$ m$^{-3}$). Caution therefore needs to be exercised when interpreting the % relative change results, especially at sites with low $NH_3$ concentrations, which must be examined together with the fitted trends.

At background sites (5 sites where total $NH_3$ emissions for the respective 5 km grid squares are estimated at <1 kg N ha$^{-1}$ y$^{-1}$), $NH_3$ concentrations also appear to have increased (non-significant). Based on the MK analysis for the period 1998 to 2014,

$NH_3$ concentrations increased overall by 18 % and 13 % from the analysis of annual and monthly data, respectively (Table 3). Results from linear regression were similar, with an overall increase of 13 % and 12 % from analysis of the annual and monthly data, respectively (Table 4). Similar to sheep sites, the % relative median change estimated from the seasonal MK Sen's slope and intercept (+ 49%) is larger than from the classic MK Sen's slope (+13%) due to differences in the intercepts of the fitted trend lines (MK = 0.1528 µg $NH_3$ m$^{-3}$ $cf$ SMK = 0.0388 µg $NH_3$ m$^{-3}$) since the trend slopes are the same (0.0012 µg $NH_3$ y$^{-1}$).

Overall, the SMK test show a significant increasing trend in the monthly data ($p = 0.05$). No individual monthly "seasons" were significant, with March, April and November monthly "seasons" approaching the significance level ($p = 0.09$).

As with the annual UK-wide long-term datasets (Sect. 3.5), it is useful to consider the significance of the $NH_3$ trends for the groupings of sites according to dominant emission source sectors. Table 3 and Table 4 shows that neither the annual nor the

monthly time series showed a significant change in $NH_3$ concentrations for the cattle dominated sites. In the case of pig and poultry dominated sites, the decrease in measured $NH_3$ concentrations was significant for both the annual and monthly datasets. For sheep dominated and backgrounds sites, the estimated increase in $NH_3$ concentrations was not significant based on the MK and linear regression tests on the annual and monthly data, but was significant based on the SMK test of the monthly data. Overall, these statistics confirm significant differences between $NH_3$ trends for sites dominated by different source types, with

concentrations decreasing at pig and poultry dominated sites, concentrations increasing at sheep dominated and background sites, and no significant trend at cattle dominated sites (Table 5).



### 3.5.6 Changing chemical climate and effects on long-term trends in $NH_3$ and $NH_4^+$

Other pollutants that affect $NH_3$ concentrations in the atmosphere include $SO_2$ and $NO_x$ emissions, which determine rates of secondary inorganic aerosol formation and therefore the lifetime of $NH_3$ in the atmosphere. UK emissions of $SO_2$ are estimated to have declined significantly by 81 % from 1.6 million tonnes in 1998 to 0.3 million tonnes in 2014 (Defra, 2015). Similarly,

$NO_x$ emissions over the same period are estimated to have fallen by 50 % from 2 million tonnes to 1 million tonnes (Defra, 2015). The reaction of $NH_3$ with $H_2SO_4$ to form $(NH_4)_2SO_4$ is effectively irreversible (in the absence of in-cloud reprocessing), whereas an equilibrium exists between gaseous $NH_3$ and particulate $NH_4NO_3$ and $NH_4Cl$ components which are appreciably volatile at ambient temperatures. A change in the particulate phase from $(NH_4)_2SO_4$ to $NH_4NO_3$ suggests that $NH_3$ will remain longer in the atmosphere, since $NH_4NO_3$ is volatile and releases $NH_3$ in warm weather.

Elsewhere, a mismatch between reported trends in emissions and measurement data have similarly been investigated. The question of the 'Ammonia Gap' in the Netherlands was debated over a number of years. There, the estimated reduction in emissions due to mitigation measures was not matched by expected decreases in measured $NH_3$ concentrations in air and/or $NH_4^+$ in precipitation (Erisman et al., 2001; Bleeker et al., 2009; van Zanten et al., 2017). Similarly in Hungary, monitored

$NH_3$ concentrations from long-term measurements did not match the estimated reduction in $NH_3$ emissions following the decline in agricultural livestock population and fertiliser usage after political changes in 1989 (Horvath and Sutton, 1998). This was subsequently attributed to a reduction in $SO_2$ emissions over the same period, increasing the atmospheric lifetime of $NH_3$ (Horvath et al., 2009).

Similar interactions are seen to be occurring in the UK based on the NAMN data, where the concurrent reduction in $SO_2$ and $NO_x$ emissions over the same period (Figure 18b) should theoretically lead to a longer atmospheric lifetime of $NH_3$, thereby increasing $NH_3$ concentrations in the UK, especially in remote areas. The interpretation of the $NH_3$ measurement data can further be aided by comparison with particulate nitrate ($NO_3^-$) and sulphate ($SO_4^{2-}$) data from the UK AGANet that are made concurrently with the NAMN $NH_3$ and $NH_4^+$ measurements at 30 sites (see Sect. 2.2). For particulate $NH_4^+$, it has already been

shown in Sect. 3.3 that this regional species has less of a relationship to the dominant $NH_3$ source sectors; trend analysis was therefore undertaken using all $NH_4^+$ site data combined. As with the $NH_3$ time series analysis, sites with incomplete data runs for particulate $NH_4^+$ due to reduced density of $NH_4^+$ measurements and site changes occurring from 2001-2006 were excluded (see Sect. 2.2.1).

**<INSERT TABLE 6>**

Two data series for NAMN $NH_4^+$ data were selected for analysis, i) 23 sites with complete $NH_4^+$ time series from 1999 to 2014, and ii) 30 sites with complete $NH_4^+$ time series from 2006 to 2014. Both time series show a large significant downward trend in $NH_4^+$ ($p < 0.01$) (Table 6, Supp. Figure S4). Overall, MK and LR tests show a significant decrease in $NH_4^+$

concentrations by 47 % and 49 %, respectively, between 1999 and 2014 and by 44 % and 43 %, respectively, between 2006 and 2014 (Table 6, Supp. Figure S4). By contrast, concurrent $NH_3$ data from the same sites over the same time periods showed a much smaller, non-significant downward trend between 1999 and 2014 ($-17$ % (MK); $-18$ % (LR)), and no discernible trend between 2006 and 2014 ($+ 3$ % (MK and LR)) (Figure 17a, Table 6, Supp. Figure S4). This reduction in particulate $NH_4^+$ can be seen to be closely associated with parallel decreases in particulate $SO_4^{2-}$ and $NO_3^-$ concentrations from AGANet (Table 7,

Figure 18a), which are themselves associated with reductions in $SO_2$ and $NO_x$ emissions (Table 7, Figure 18b).

**<INSERT FIGURE 17>**





**\<INSERT FIGURE 18\>**

**\<INSERT TABLE 7\>**

The comparisons above therefore suggest that reductions in $SO_2$ and $NO_x$ emissions over the period have led to a slower

formation of particulate $NH_4^+$ in the atmosphere. Further evidence in support of this is indicated by plotting the ratio of $NH_3/NH_4^+$ (Figure 17b), which has increased from 1.8 in 1999 to 2.8 in 2014. This demonstrates how a larger fraction of the reduced N is staying in the gas phase as $NH_3$, increasing its atmospheric residence time and maintaining $NH_3$ concentrations at a higher level than solely based on $NH_3$ emission trends. Although the overall changes in $NH_3$ concentrations in the UK dataset are small and in many cases not significant for particular data groupings, they are consistent with similar phenomena

observed in Hungary, the Netherlands and Denmark (Horvath et al., 2009; Erisman et al., 2001; Sutton et al., 2003; Bleeker et al., 2009).

**4. CONCLUSIONS**

Spatial and temporal trends in $NH_3$ are found to be related to variability in emission source types across the UK and also to be influenced by changes in environmental conditions. Extensive spatial heterogeneity in $NH_3$ concentrations was observed, with lowest annual mean concentrations at remote sites (< 0.2 µg m$^{-3}$) and highest in the areas with intensive agriculture (up to 22 µg m$^{-3}$). $NH_4^+$ concentrations show less spatial variability (e.g. range of 0.14 to 1.8 µg m$^{-3}$ annual mean in 2005) with a general decreasing gradient from the south-east to the north-west of the UK, due to both regional differences in $NH_3$ concentrations

and import of particulate matter into south-east England from Europe (Vieno et al., 2014; Dore et al., 2015).

Peak $NH_3$ concentrations are observed in summer at background sites (defined by 5 km grid average $NH_3$ emissions <1 kg N ha$^{-1}$ y$^{-1}$) and in areas dominated by sheep farming, driven by increased volatilization of $NH_3$ in warmer summer temperatures. In areas where cattle, pig and poultry farming is dominant, the largest $NH_3$ concentrations are in spring and autumn, matching

periods of manure application to fields. By contrast, peak concentrations of $NH_4^+$ aerosol occur in spring from long-range transboundary sources. The spatial and seasonal patterns established for sites influenced by different emission source sectors are important for providing a foundation to understanding $NH_3$ exchange processes, impacts and the UK $NH_3$ budget, and to inform abatement strategies.

Official published estimates of UK $NH_3$ emissions are estimated to have declined by 16.3 % between 1998 and 2014. The long-term $NH_3$ concentration data from the UK NAMN suggests evidence of a smaller, but non-significant decreasing trend (−6.3 % (MK); −3.1 % (LR)), based on analysis of annually averaged data ($n = 59$) over the same period (Table 2). Analysis of annually averaged data for different groupings of the NAMN dataset for the time periods 1999-2014 ($n = 66$) and 2000-2014 ($n = 75$) also gave similar results. In each case, the level of reduction observed in the datasets (1999-2014: 0.0 % (MK)

*vs* −3.0 % (LR); 2000-2014: 0.0 % (MK) *vs* −2.8 % (LR)) is less than the 15.6 % and 13.1 % reduction in estimated UK $NH_3$ emissions over the periods 1999-2014 and 2000-2014, respectively (Table 2).

In areas with intensive pig and poultry farming, there is a significant downward trend in $NH_3$ concentrations from the analysis of annually averaged data (−22 % (MK), $p = 0.02$; −21 % (LR), $p = 0.06$) that is consistent with, but not as large as the decrease

in estimated $NH_3$ emissions from this sector over the same period (−39 %) (Table 5). By contrast, in cattle-dominated areas, there is evidence of a small increasing, but non-significant trend in $NH_3$ concentrations (+12 % (MK); +3.6 % (LR): annually averaged data), despite the decline in $NH_3$ emissions from this sector since 1998 (−11%) (Table 5). At background and sheep





dominated sites, $NH_3$ concentrations increased (non-significant) over the monitoring period (Table 5). These increases in $NH_3$ concentrations at background (+17 % (MK); +13 % (LR): annually averaged data) and sheep dominated sites (+15 % (MK); +19 % (LR): annually averaged data) are consistent with decreasing $SO_2$ emissions (and to a lesser extent $NO_x$ emissions) associated with a change in the PM from $(NH_4)_2SO_4$ to $NH_4NO_3$, the latter being volatile and releasing $NH_3$ in warm weather.

Particulate $NH_4^+$ represents a secondary pollutant formed from $NH_3$ and oxidation products of acidic gases such as $SO_2$ and $NO_x$. As the emissions of these acidic gases have reduced over the past years, the ratio between $NH_3$ and $NH_4^+$ has increased from 1.8 to 2.8 between 1999 and 2014. These changes are consistent with observed decreases in particulate $SO_4^{2-}$ and $NO_3^-$ concentrations that are associated with decline in $SO_2$ and $NO_x$ emissions over the same period. This effect appears to be of
10  sufficient magnitude to explain the lack of overall decrease in $NH_3$ concentrations, where the decrease in $NH_4^+$ is larger than for $NH_3$ at corresponding sites.  Overall, UK annual particulate $NH_4^+$ concentrations decreased by −47 % (MK) and −49 % (LR) for period 1999 -2014, associated with a slower formation of particulate $NH_4^+$ in the atmosphere from gas-phase $NH_3$. The findings are consistent with a parallel change in partitioning from particulate $NH_4^+$ to gaseous $NH_3$ as also detected in Hungary, the Netherlands and Denmark.

Until now, only a modest commitment has been agreed to reduce European $NH_3$ emissions. By contrast, $SO_2$ and $NO_x$ emissions have decreased over Europe over the past decades, and are projected to decrease further under the revised Gothenburg Protocol and revised NECD. As a result, the importance of $NH_3$ relative to oxidised N and $SO_2$ emissions is expected to continue to increase over the next decades, playing a significant role in the formation of fine PM and contributing
20  to ecosystem effects through N deposition. With longer atmospheric lifetimes of gaseous $NH_3$ and little commitment to reduce emissions, combined with climate warming effects tending to increase $NH_3$ emissions, there is a substantial risk that exceedance of the $NH_3$ critical levels may increase in the future, exacerbating the threat to the most sensitive semi-natural habitats. The growing relative importance of reduced nitrogen to total acidic and total nitrogen deposition indicates that future strategies to tackle acidification and eutrophication will need to include measures to abate emissions of $NH_3$.

**Acknowledgements**
This work was carried out with funding from the Department for Environment, Food and Rural Affairs (Defra) and the devolved administrations, and from supporting NERC CEH programmes. The assistance and contributions from the large
30  network of NAMN and AGANet site operators, former NAMN network managers (Ben Miners, Antje Branding), the Centralised Analytical Chemistry in Lancaster and in particular Heather Carter, Darren Sleep and Philip Rowland, colleagues at CEH Edinburgh (Sarah Leeson, Matt Jones, Chris Andrews, Margaret Anderson, David Leaver), and Ricardo Energy & Environment (Martin Davies, Tim Bevington, Ben Davies) is also gratefully acknowledged.



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




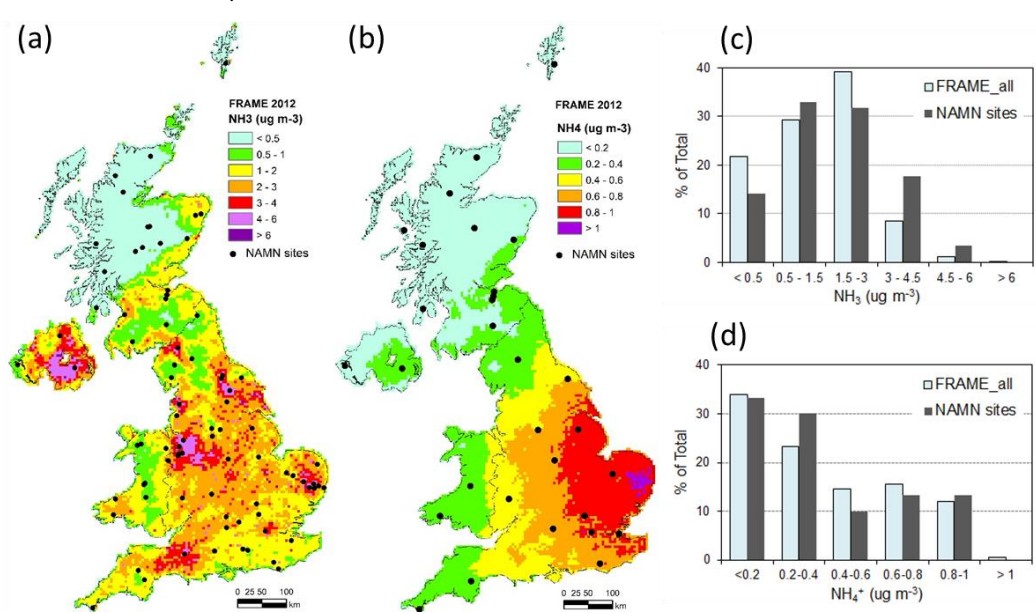

**Figure 1: Maps of modelled annual mean concentrations of (a) $NH_3$ and (b) $NH_4^+$ at 5 km × 5 km grid resolution from the FRAME atmospheric transport model using 2012 UK emissions data, based on Dore et al. (2008), overlaid with the National Ammonia Monitoring Network (NAMN) measurement sites, and frequency distributions of the modelled concentrations of (c) $NH_3$ and (d) $NH_4^+$ for the FRAME 5 km grid squares containing a NAMN site (85 and 30 sites, respectively, in 2012) and for all model grid squares over the UK.**

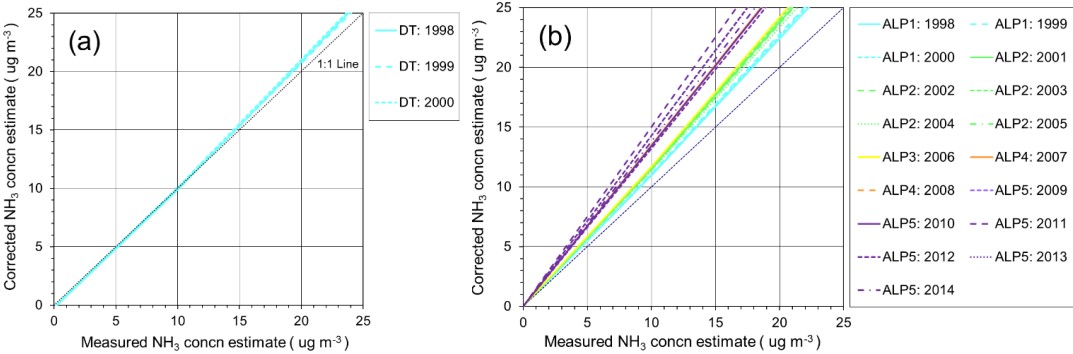

**Figure 2: Comparison of annual empirical calibration curves for the passive samplers against the reference estimates from DELTA sampling at > 8 sites in the UK National Ammonia Monitoring Network (NAMN). (a) DT = Diffusion Tubes. (b) ALP = ALPHA samplers, ALP1 is prototype 1 (1998-2000), ALP2 (2001-2005) and ALP3-ALP5 were manufactured from injection moulds 1 and 2, respectively. ALP4 and ALP 5 = new inlet PTFE membrane (Swiftlab 07-OPM-027: 305 μm, regular polypropylene grid support material) that replaced the previous TE38 PTFE membrane (265 μm, randomly arranged polypropylene support material). ALP5 = new laboratory with analysis on FloRRia (previously on AMFIA).**





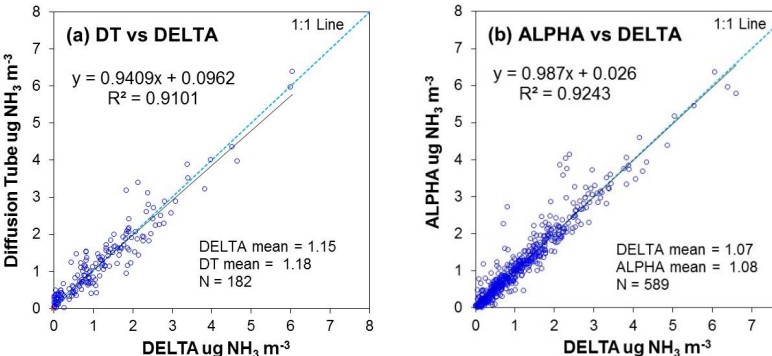

**Figure 3: Regression of passive samplers vs DELTA measurements at >8 sites in the UK National Ammonia Monitoring Network (NAMN), showing results for (a) diffusion tubes (DT), used during the early years of the network (1998-2000), and (b) for ALPHA samplers (results shown are for 2009-2014 where all analyses were carried out at a new laboratory). All passive data shown are the monthly measured concentrations for each site using the calibrated data for the respective passive methods.**

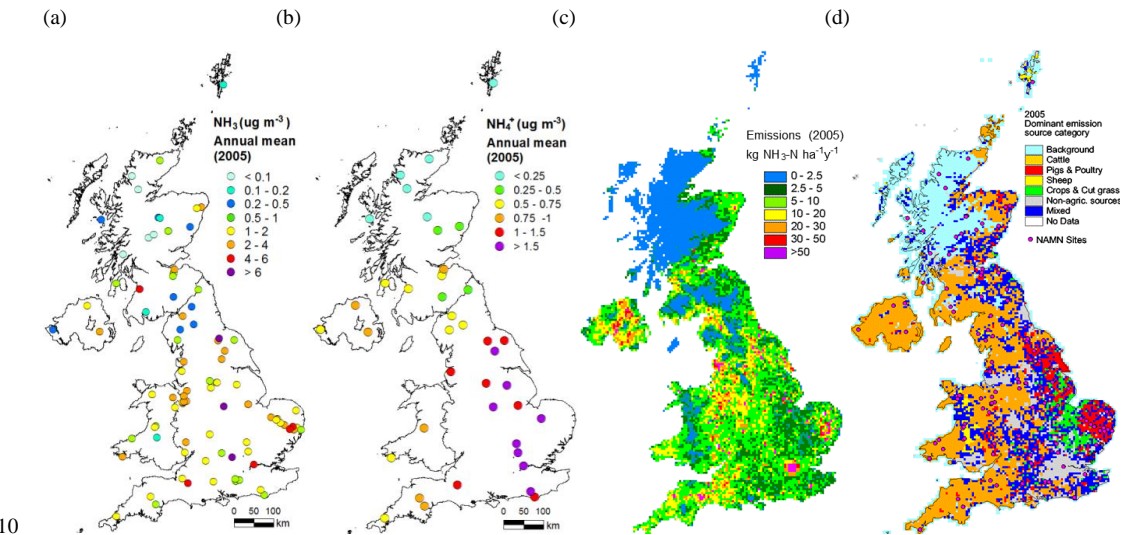

**Figure 4: Measured annual mean concentrations from the UK National Ammonia Monitoring Network (NAMN) for 2005 for (a) NH₃ and (b) particulate NH₄⁺, and maps at 5 km by 5 km grid resolution for 2005 of (c) the estimated annual NH₃ emissions (Dragosits et al. 2005) and (d) the dominant NH₃ emission source category (based on Hellsten et al., 2008), indicating the relationships between measured air concentrations and spatial variability in NH₃ sources emissions.**





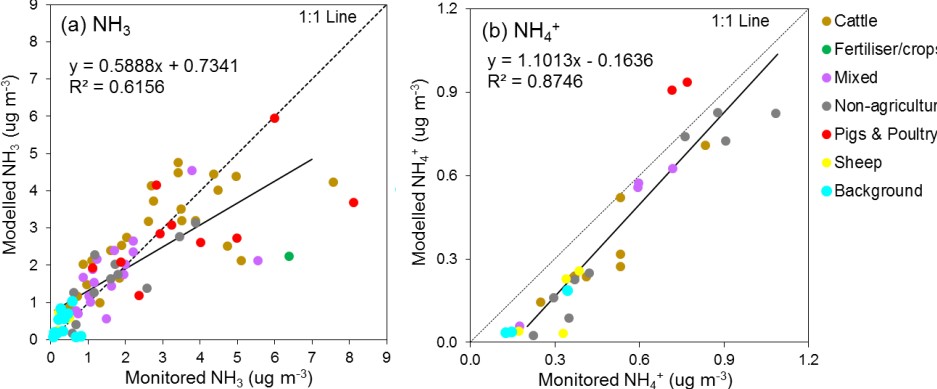

**Figure 5: Comparison of 2012 annual mean concentrations of (a) NH₃ and (b) NH₄⁺ modelled using the FRAME atmospheric model with 2012 measurements from the UK National Ammonia Monitoring Network (NAMN) for all sites according to dominant emission**
5  **source classification.**

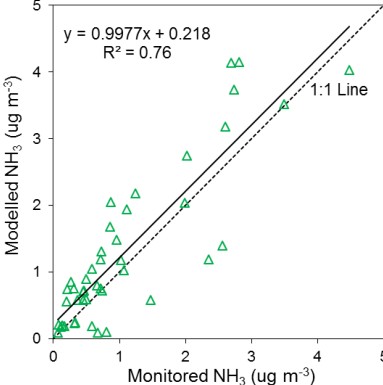

**Figure 6: Comparison of 2012 annual mean concentrations of NH₃ from output of the FRAME atmospheric model with measurements from the UK National Ammonia Monitoring Network (NAMN) for a subset of sites classified as located in semi-natural or forest locations.**



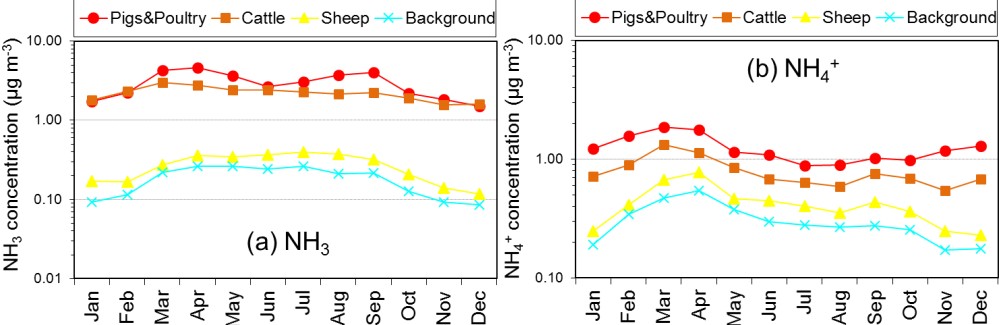

**Figure 7: Seasonal trends in (a) NH₃ (mean monthly data for 1998-2014) and (b) NH₄⁺ (mean monthly data for 1999-2014) concentrations of sites in the UK National Ammonia Monitoring Network (NAMN) classified according to four key emission source categories: cattle, sheep, pigs & poultry and background (based on 2005 dominant emission source classification). The concentrations are plotted on a log scale for better visualisation of the low concentration background and sheep profiles.**

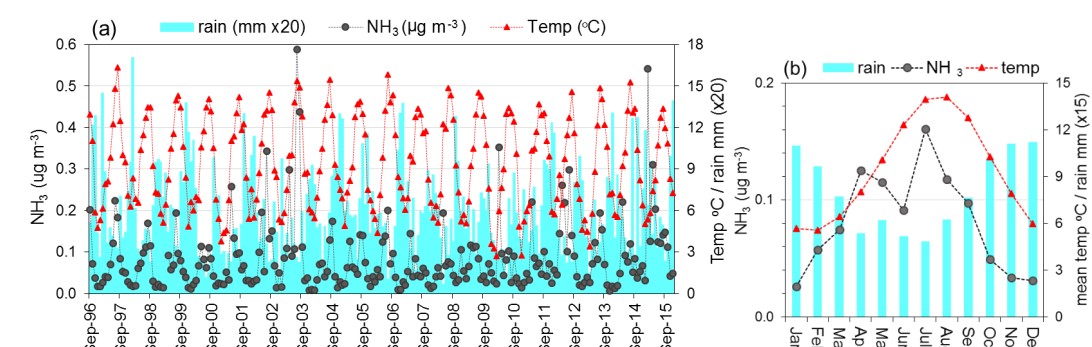

**Figure 8: (a) Long-term trends in measured monthly-mean NH₃ concentrations at the remote background Inverpolly site in NW Scotland (UKA00457), demonstrating strong intra- and inter-annual variability, from the UK National Ammonia Monitoring Network (NAMN). Also plotted for comparison are monthly rainfall and temperature data from the nearby Aultbea meteorological station (ID no. 52; MetOffice, 2016). (b) Comparison of seasonal trends in NH₃ concentrations with temperature and rainfall at Inverpolly. Data shown are averaged over the period 1996 – 2015. Peak concentrations of NH₃ can be seen to coincide with summer maxima in the temperature profile, while the lowest concentration occur in winter when the temperature is lowest and also when rainfall is generally highest.**





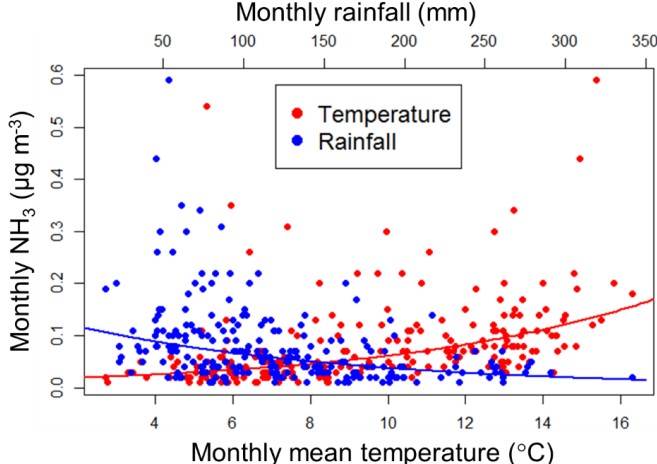

**Figure 9. Relationships between measured monthly-mean NH₃ concentrations from the UK National Ammonia Monitoring Network (NAMN) and mean monthly temperature and rainfall at Inverpolly (UKA00457). NH₃ was negatively correlated with rainfall (blue line: $Log(NH_3) = -0.0059*Log(rain) - 2.1612$, $R^2 = 0.19$, $n = 231$, $p < 0.05$) and positively correlated with temperature (red line: $Log(NH_3) = 0.1482*Log(temp) - 4.2708$ $R^2 = 0.33$, $n = 231$, $p < 0.05$). Rain and temperature data are from the nearby Aultbea meteorological station (ID no. 52; MetOffice, 2016).**

(a)                     (b)

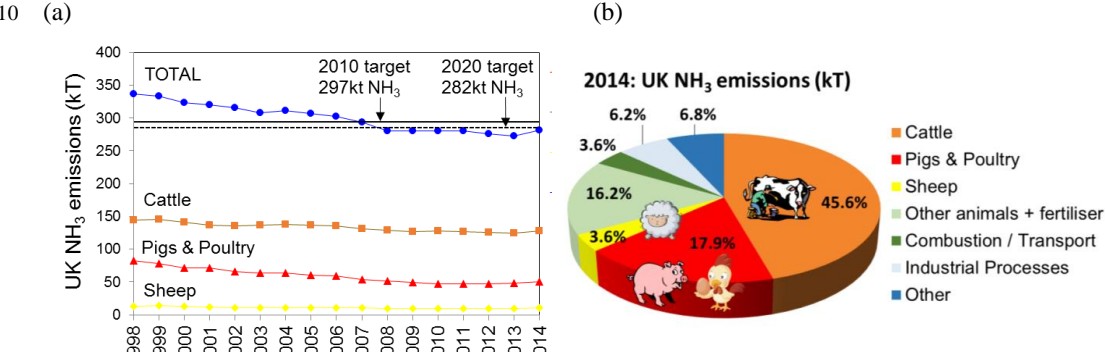

**Figure 10: (a) Trends between 1998 and 2014 in the UK National Atmospheric Emission Inventory (NAEI) for total UK NH₃ emissions and selected sub-sources: cattle, pigs & poultry and sheep. The 2010 NH₃ national emissions ceilings target of 297 kt (Gothenburg protocol and NECD) and the 2020 target of 282 kt (revised Gothenburg protocol) area are also shown for comparison. (b) UK NH₃ emission sources in 2014. Data from *http://naei.defra.gov.uk/* and Misselbrook et al. 2015.**





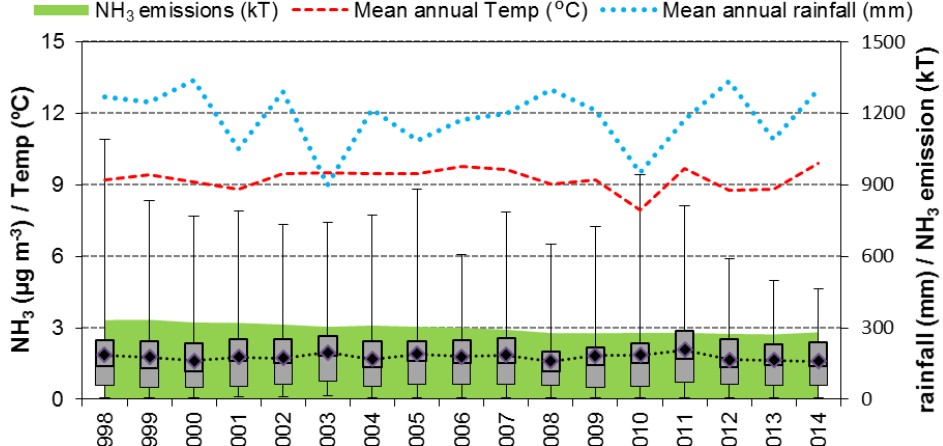

**Figure 11: Changes in annual mean atmospheric NH₃ concentrations averaged over all sites in the National Ammonia Monitoring Network (NAMN) operational between 1998 and 2014 (59 sites). The diamonds show the mean NH₃ concentration, with the grey box indicating the median and interquartile range, while the error bars show the range (minimum and maximum) of measured mean concentrations. Annual mean UK meteorological data (source _http://www.metoffice.gov.uk/_) are also plotted for comparison over the same period. 2010 was an unusual year, characterised by a considerably lower than average mean annual temperature of 7.9 °C due to exceptionally cold winters, with Dec 2010 recorded as the coldest for over 100 years (_cf._ mean = 9.2 °C for 1998 to 2014) and lower than average rainfall of 950 mm (_cf_ mean = 1190 mm for 1998 to 2014).**





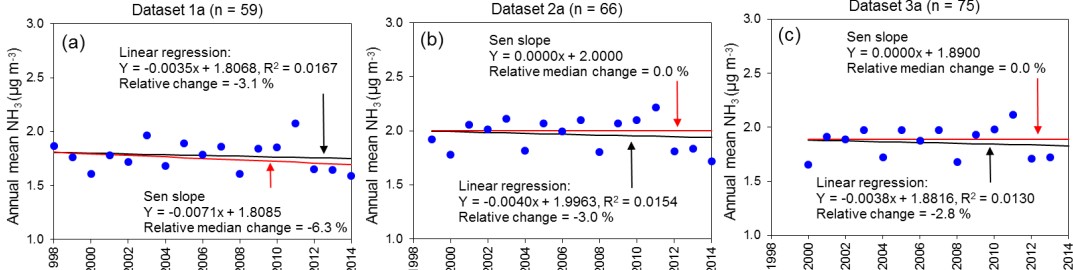

**Figure 12: Time series trend analysis by non-parametric Mann-Kendall Sen slope *vs* parametric linear regression on annually averaged NH₃ concentrations from the UK National Ammonia Monitoring Network (NAMN) for a) dataset 1a (1998-2014, *n*=59), b) dataset 2a (1999-2014, *n*= 66) and c) dataset 3a (2000-2014, *n*=75). Individual data points are annually averaged NH₃ concentrations.**

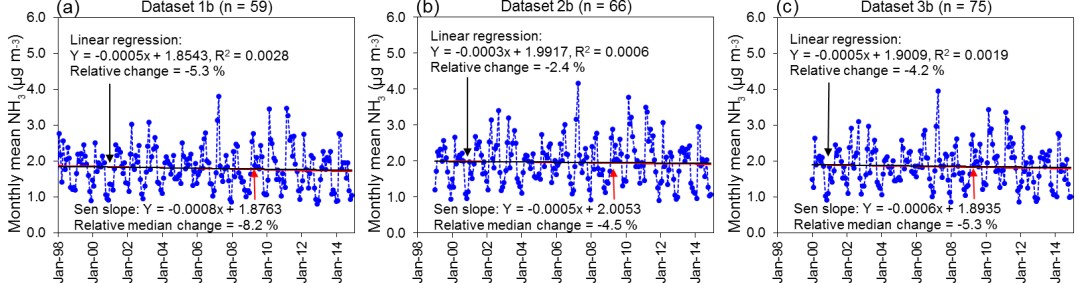

**Figure 13: Time series trend analysis by non-parametric Mann-Kendall Sen slope *vs* parametric linear regression on monthly mean NH₃ concentrations from the UK National Ammonia Monitoring Network (NAMN) for a) dataset 1b (1998-2014, *n*=59), b) dataset 2b (1999-2014, *n*= 66) and c) dataset 3b (2000-2014, *n*=75). Individual data points are monthly mean NH₃ concentrations.**


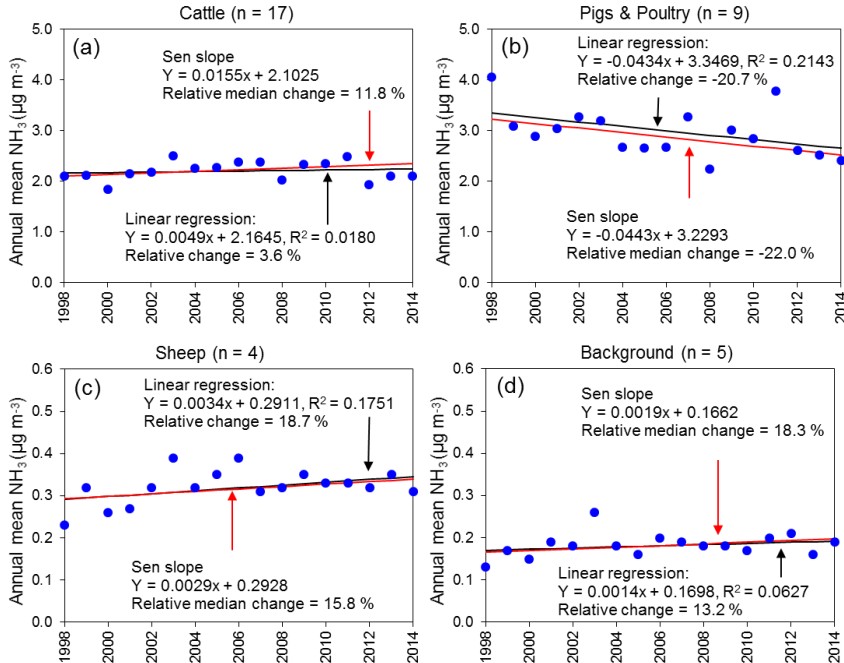

**Figure 14: Time series trend analysis by non-parametric Mann-Kendall Sen slope *vs* parametric linear regression on annually averaged NH₃ concentrations from the UK National Ammonia Monitoring Network (NAMN) for sites in 5 km grid squares classed as dominated by (a) cattle (> 45 % of total NH₃ emissions from this category in a grid square); (b) pigs & poultry (> 45 % of total NH₃ emissions from this category in a grid square); (c) sheep (> 45 % of total NH₃ emissions from sheep in a grid square); (d) NAMN sites in grid squares classed as background (defined as grid squares with average NH₃ emissions <1 kg N ha⁻¹ y⁻¹). Individual data points are annually averaged NH₃ concentrations.**





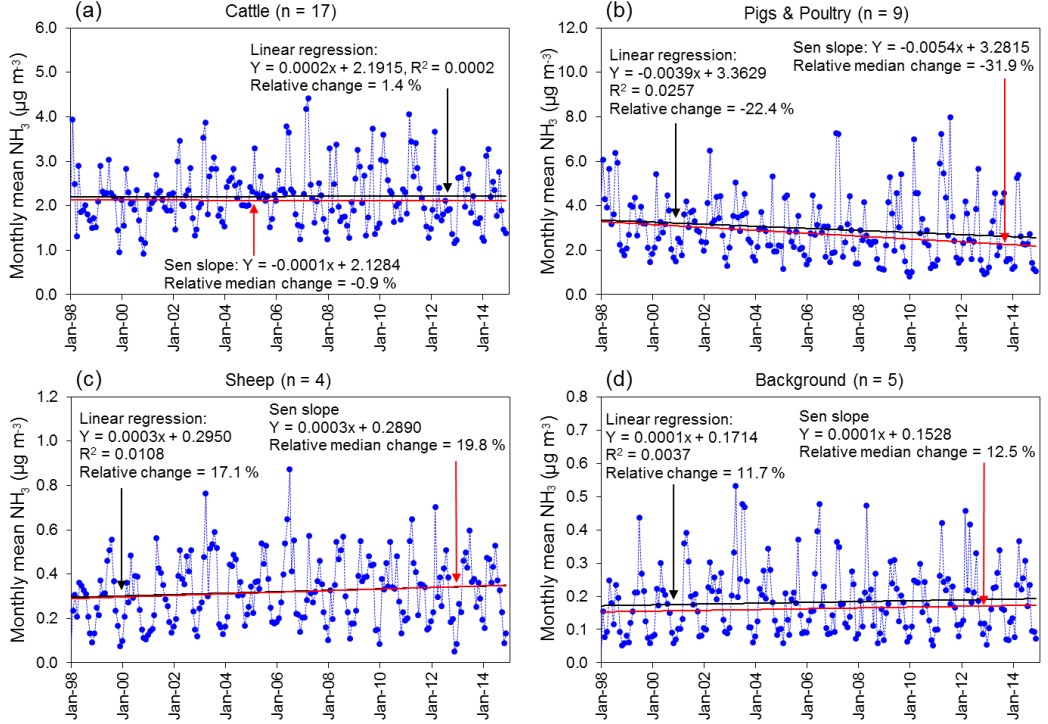

**Figure 15: Time series trend analysis by non-parametric Mann-Kendall Sen slope *vs* parametric least squares linear regression on**
5  **annually averaged NH$_3$ concentrations from the UK National Ammonia Monitoring Network (NAMN) for sites in 5 km grid squares**
**classed as dominated by (a) cattle (> 45 % of total NH$_3$ emissions from this category in a grid square); (b) pigs & poultry (> 45 % of**
**total NH$_3$ emissions from this category in a grid square); (c) sheep (> 45 % of total NH$_3$ emissions from sheep in a grid square); (d)**
**NAMN sites in grid squares classed as background (defined as grid squares with average NH$_3$ emissions <1 kg N ha$^{-1}$ y$^{-1}$). Individual**
**data points are monthly mean NH$_3$ concentrations.**




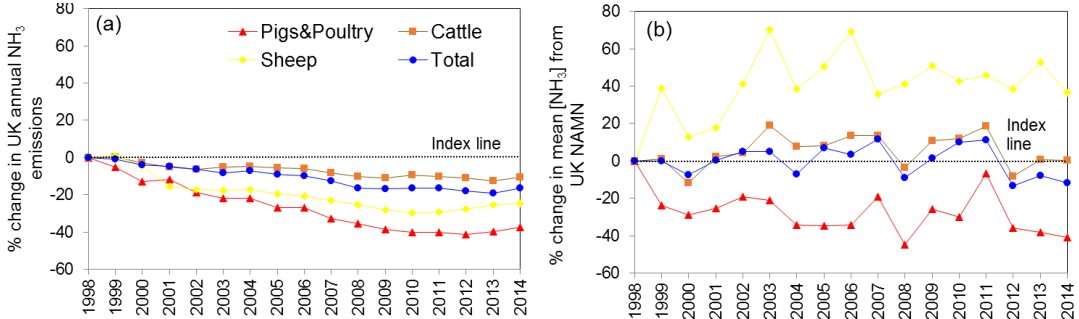

**Figure 16: (a) Relative trends between 1998 and 2014 in NH₃ emissions from the UK National Atmospheric Emission Inventory**
**(NAEI) for total emissions (all NH₃ sources) and emissions from cattle, pigs & poultry, and sheep separately (data from:**
***http://naei.defra.gov.uk/* and Misselbrook et al, 2015). (b) Relative trends between 1998 and 2014 in measured annual mean NH₃**
**concentrations for all UK National Ammonia Monitoring Network (NAMN) sites, and for sites classified as dominated by cattle, pigs**
**& poultry, and sheep. Both figures are plotted with the same scale to allow direct comparison of the relative magnitudes in trends.**

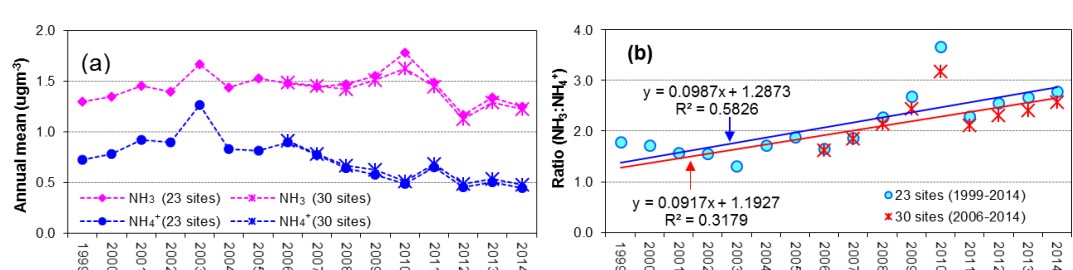

**Figure 17: (a) Long-term trends in annual mean concentrations of measured particulate NH₄⁺ from the UK National Ammonia**
**Monitoring Network (NAMN) comparing i) 23 sites with complete NH₄⁺ time series from 1999 to 2014, and ii) 30 sites with complete**
**NH₄⁺ time series from 2006 to 2014. For comparison, long-term trends in annual mean concentrations of measured NH₃ from the**
**same sites over the same time periods are also shown. (b) Plot of ratio of NH₃:NH₄⁺, based on data from graph a, indicating an**
**increase in this ratio with time.**

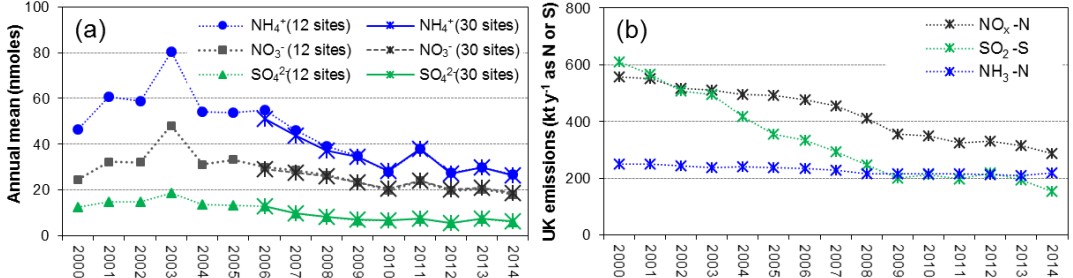

**Figure 18: (a) Long-term trends in particulate NH₄⁺ from the UK National Ammonia Monitoring Network (NAMN) compared with**
**particulate NO₃⁻ and SO₄²⁻ concentrations from the UK Acid Gas and Aerosol Network (AGANet; Connolly et al. 2016) measured at**
**the same time. Each data point represents the averaged monthly measurements from all AGANet sites (increased from 12 to 30 sites**
**since Jan 2006) and also the original l2 AGANet sites in the network (1999 data were excluded as measurements started in September**
**1999). (b) Trends in total UK emissions of NH₃, NOₓ and SO₂ over the same period (2000-2014). Data from the National Atmospheric**
**Emission Inventory (NAEI, *http://naei.defra.gov.uk/*).**





**Table 1: Summary of Mann-Kendall (MK) and Seasonal Mann-Kendall (SMK) time series trend analysis on NH₃ data (annually averaged datasets 1a, 2a, 3a and monthly mean datasets 1b, 2b, 3b) from the UK National Ammonia Monitoring Network (NAMN). The following are shown: the *p*-value, median annual trend (Sen's slope, in µg NH₃ y⁻¹) and the relative median change over the selected time period (in %). For the MK tests, the 95% confidence interval (CI) for the trend and relative change are also estimated.**
**For comparison, the reduction in estimated UK NH₃ emissions over the periods 1998-2014, 1999-2014 and 2000-2014 are 16.3 %, 15.6 % and 13.1 % respectively.**

| Dataset | Time series | [a]Number of sites | *p-value* | Significant trend ($p<0.05$) | [b]Median annual trend & [95% CI] (µg NH₃ y⁻¹) | [c]Relative median change over the period & [95% CI] (%) |
|---|---|---|---|---|---|---|
| 1a: annual (MK) | 1998-2014 | 59 | 0.46 | no | -0.0071 [-0.0200, 0.0125] | -6.3 [-16, 12] |
| 1b: monthly (MK) | 1998-2014 | 59 | 0.22 | no | -0.0096 [-0.0264, 0.0060] | -8.2 [-21, 5.5] |
| 1b: monthly (SMK) | 1998-2014 | 59 | 0.10 | no | -0.0100 | -5.8 |
| 2a: annual (MK) | 1999-2014 | 66 | 1.00 | no | 0.0000 [-0.0227, 0.0200] | 0.0 [-16, 16] |
| 2b: monthly (MK) | 1999-2014 | 66 | 0.51 | no | -0.0060 [-0.0252, 0.0132] | -4.5 [-18, 11] |
| 2b: monthly (SMK) | 1999-2014 | 66 | 0.25 | no | -0.0073 | -4.2 |
| 3a: annual (MK) | 2000-2014 | 75 | 1.00 | no | 0.0000 [-0.0283, 0.0175] | 0.0 [-19, 14] |
| 3b: monthly (MK) | 2000-2014 | 75 | 0.43 | no | -0.0072 [-0.0264, 0.0120] | -5.3 [-18, 9.5] |
| 3b: monthly (SMK) | 2000-2014 | 75 | 0.15 | no | -0.0079 | -4.5 |

[a]Number of sites providing complete data runs over the time period.
[b]Median annual trend = fitted Sen's slope of Mann-Kendall linear trend (unit = µg NH₃ y⁻¹)
[c]Relative median change calculated based on the NH₃ concentration at the start ($y_0$) and at the end ($y_i$) of time series computed
from the Sen's slope and intercept ($=100*[(y_i-y_0)/y_0]$).

**Table 2: Summary of linear regression time series trend analysis on NH₃ data (annually averaged datasets 1a, 2a, 3a and monthly**
**mean datasets 1b, 2b, 3b) from the UK National Ammonia Monitoring Network (NAMN). The following are shown: the *p*-value, annual trend (fitted slope, in µg NH₃ y⁻¹), $R^2$, and the relative change over the selected time period (in %). For comparison, the reduction in estimated UK NH₃ emissions over the periods 1998-2014, 1999-2014 and 2000-2014 are 16.3 %, 15.6 % and 13.1 % respectively.**

| Dataset | Time series | [a]Number of sites | *p-value* | Significant trend ($p<0.05$) | [b]Annual Trend (µg NH₃ y⁻¹) | $R^2$ | [c]Relative change over the period (%) |
|---|---|---|---|---|---|---|---|
| 1a: annual | 1998-2014 | 59 | 0.62 | no | -0.0035 | 0.0167 | -3.1 |
| 1b: monthly | 1998-2014 | 59 | 0.45 | no | -0.0062 | 0.0028 | -5.3 |
| 2a: annual | 1999-2014 | 66 | 0.65 | no | -0.0040 | 0.0154 | -3.0 |
| 2b: monthly | 1999-2014 | 66 | 0.74 | no | -0.0031 | 0.0006 | -2.4 |
| 3a: annual | 2000-2014 | 75 | 0.69 | no | -0.0038 | 0.0130 | -2.8 |
| 3b: monthly | 2000-2014 | 75 | 0.56 | no | -0.0057 | 0.0019 | -4.2 |

[a]Number of sites providing complete data runs over the time period.
[b]Annual trend = fitted slope of linear regression (unit = µg NH₃ y⁻¹)
[c]Relative change calculated based on the estimated annual NH₃ concentration at the start ($y_0$) and at the end ($y_i$) of time series ($=100*[(y_i-y_0)/y_0]$) computed from the slope and intercept ($=100*[(y_i-y_0)/y_0]$).



**Table 3: Summary of Mann-Kendall (MK) and Seasonal Mann-Kendall (SMK) time series trend analysis on grouped NH₃ concentration data (annually averaged and monthly mean data) from the UK National Ammonia Monitoring Network (NAMN) for four different emission source sectors. The following are shown: the *p*-value, median annual trend (Sen's slope, in µg NH₃ y⁻¹) and the relative median change over the selected time period (in %). For the MK tests, the 95% confidence interval (CI) for the trend and relative change are also estimated.**

| Source sector | Time series (1998-2014) | [a]Number of sites | *p*-value | Significant trend (*p*<0.05) | [b]Median annual trend & [95% CI] (µg NH₃ y⁻¹) | [c]Relative median change over the period & [95% CI] (%) |
|---|---|---|---|---|---|---|
| Cattle | Annual (MK) | 17 | 0.46 | no | 0.0155 [-0.0150, 0.0300] | 12 [-10, 24] |
| Cattle | Monthly (MK) | 17 | 0.90 | no | -0.0012 [-0.0192, 0.0168] | -0.9 [-14, 13] |
| Cattle | Monthly (SMK) | 17 | 0.51 | no | 0.0043 | 3.9 |
| Pigs&Poultry | Annual (MK) | 9 | 0.02 | yes | -0.0043 [-0.1008,-0.0071] | -22 [-42, -3.9] |
| Pigs&Poultry | Monthly (MK) | 9 | < 0.001 | yes | -0.0648 [-0.0984,-0.0300] | -32 [-46, -16] |
| Pigs&Poultry | Monthly (SMK) | 9 | < 0.001 | yes | -0.0588 | -11 |
| Sheep | Annual (MK) | 4 | 0.17 | no | 0.0029 [0.0000, 0.0069] | 16 [0.0, 46] |
| Sheep | Monthly (MK) | 4 | 0.10 | no | 0.0036 [0.0000, 0.0072] | 20 [0.0, 45] |
| Sheep | Monthly (SMK) | 4 | < 0.01 | yes | 0.0033 | 210 |
| Background | Annual (MK) | 5 | 0.20 | no | 0.0019 [-0.0012, 0.0038] | 18 [-10, 41] |
| Background | Monthly (MK) | 5 | 0.23 | no | 0.0012 [-0.0012, 0.0036] | 13 [-11, 42] |
| Background | Monthly (SMK) | 5 | 0.05 | yes | 0.0012 | 49 |

[a]Number of sites providing complete data runs over the period 1998 to 2014.

[b]Median annual trend = fitted Sen's slope of Mann-Kendall linear trend (unit = µg NH₃ y⁻¹)

[c]Relative median change calculated based on the annual NH₃ concentration at the start ($y_0$) and at the end ($y_i$) of time series computed from the Sen's slope and intercept (=100*[($y_i$-$y_0$)/$y_0$]).

*Cattle sites*: Bickerton Hill (UKA00297), Brown Moss (UKA00369), Castle Cary (UKA00328), Cwmystwyth (UKA00325), Fenn's Moss (UKA00291), High Muffles (UKA00169), Hillsborough (UKA00293), Little Budworth (UKA00298), Llynclys Common (UKA00270), Lough Navar (UKA00166), Myerscough (UKA00356), Northallerton (UKA00316), North Wyke (UKA00269), Penallt (UKA00324), Wardlow Hay Cop (UKA00119), Wem Moss (UKA00299), Yarner Wood (UKA00168).

*Pig & Poultry sites*: Bedlingfield (UKA00334), Dennington (UKA00331), Dunwich Heath (UKA00308), Fressingfield (UKA00335), Mere Sands Wood (UKA00280), Redgrave + Lopham (UKA00311), Sibton (UKA00012), Stoke Ferry (UKA00317), Stanford (UKA00476).

*Sheep sites*: Glensaugh (UKA00348; 2005 classification = background, but 1km radius is predominantly sheep from local landuse information), Moorhouse (UKA00357) and Sourhope (UKA00347) (2015 classification = cattle, but 1km radius around site is sheep from local landuse information), (Shetland UKA00486).

*Background sites*: Allt a Mharcaidh (UKA00086), Dumfries (UKA00368), Eskdalemuir (UKA00130), Inverpolly (UKA00457), Strathvaich (UKA00162).





Table 4: Summary of linear regression time series trend analysis on grouped $NH_3$ concentration data (annually averaged data and also monthly mean data) from the UK National Ammonia Monitoring Network (NAMN) for four different emission source sectors. The following are shown: the *p*-value, annual trend (fitted slope, in µg $NH_3$ y$^{-1}$), $R^2$, and the relative change over the selected time period (in %).

| Source sector | Time series (1998-2014) | [a]Number of sites | *p-value* | Significant trend ($p<0.05$) | [b]Annual Trend (µg $NH_3$ y$^{-1}$) | $R^2$ | [b] Relative change over the period [%] |
|---|---|---|---|---|---|---|---|
| Cattle | annual | 17 | 0.61 | no | 0.0049 | 0.0180 | 3.6 |
| Cattle | monthly | 17 | 0.84 | no | 0.0019 | 0.0002 | 1.4 |
| Pigs&Poultry | annual | 9 | 0.06 | no | -0.0434 | 0.2143 | -21 |
| Pigs&Poultry | monthly | 9 | 0.02 | yes | -0.0466 | 0.0257 | -22 |
| Sheep | annual | 4 | 0.09 | no | 0.0034 | 0.1751 | 19 |
| Sheep | monthly | 4 | 0.14 | no | 0.0032 | 0.0108 | 17 |
| Background | annual | 5 | 0.33 | no | 0.0014 | 0.0627 | 13 |
| Background | monthly | 5 | 0.39 | no | 0.0013 | 0.0037 | 12 |

5   [a]Number of sites providing complete data runs over the specified time period in analysis

[b]Annual trend = fitted slope of linear regression (unit = µg $NH_3$ y$^{-1}$)

[c]Relative change calculated based on the estimated annual $NH_3$ concentration at the start ($y_0$) and at the end ($y_i$) of time series (=100*[($y_i$-$y_0$) /$y_0$]) computed from the slope and intercept (=100*[($y_i$-$y_0$) /$y_0$]).

10   *Cattle sites*: Bickerton Hill (UKA00297), Brown Moss (UKA00369), Castle Cary (UKA00328), Cwmystwyth (UKA00325), Fenn's Moss (UKA00291), High Muffles (UKA00169), Hillsborough (UKA00293), Little Budworth (UKA00298), Llynclys Common (UKA00270), Lough Navar (UKA00166), Myerscough (UKA00356), Northallerton (UKA00316), North Wyke (UKA00269), Penallt (UKA00324), Wardlow Hay Cop (UKA00119), Wem Moss (UKA00299), Yarner Wood (UKA00168).

*Pig & Poultry sites*: Bedlingfield (UKA00334), Dennington (UKA00331), Dunwich Heath (UKA00308), Fressingfield (UKA00335), Mere 15   Sands Wood (UKA00280), Redgrave + Lopham (UKA00311), Sibton (UKA00012), Stoke Ferry (UKA00317), Stanford (UKA00476).

*Sheep sites*: Glensaugh (UKA00348; 2005 classification = background, but 1km radius is predominantly sheep from local landuse information), Moorhouse (UKA00357) and Sourhope (UKA00347) (2015 classification = cattle, but 1km radius around site is sheep from local landuse information), (Shetland UKA00486).

*Background sites*: Allt a Mharcaidh (UKA00086), Dumfries (UKA00368), Eskdalemuir (UKA00130), Inverpolly (UKA00457), Strathvaich 20   (UKA00162).





**Table 5: Comparison of % change in estimated UK NH₃ emissions reported by the National Atmospheric Emission Inventory (NAEI) (data from*: http://naei.defra.gov.uk/*) with % change between 1998 and 2014 in annually averaged NH₃ concentration data from the UK National Ammonia Monitoring Network (NAMN) for all NAMN sites (dataset 1a) and for grouped sites in four different emission source sectors.**

| Comparison period: 1998 - 2014 | All sites (dataset 1a: n = 59) | Cattle (n=17) | Pigs & Poultry (n=9) | Sheep (n=4) | Background (n=5) |
|---|---|---|---|---|---|
| UK NH₃ emissions: % change relative to 1998 | -16 | -11 | -39 | -24 | no data |
| UK NAMN NH₃: % relative median change estimated from MK Sen's slope and intercept | -6.3 (see Table 1) | 12 (see Table 3) | -22* (see Table 3) | 15 (see Table 3) | 17 (see Table 3) |
| UK NAMN NH₃: % relative change estimated from linear regression slope and intercept | -3.1 (see Table 2) | 3.6 (see Table 4) | -21^Δ (see Table 4) | 19 (see Table 4) | 13 (see table 4) |

5  Significance: *$p < 0.05$, **$p < 0.01$, ***$p < 0.001$, ^Δ$p = 0.06$.

**Table 6: Comparison of % change in UK NH₃, SO₂ and NO$_x$ emissions reported by the National Atmospheric Emission Inventory (NAEI) (data from*: http://naei.defra.gov.uk/*) with % change in annually averaged NH₄⁺ and NH₃ concentration data from the UK**
10 **National Ammonia Monitoring Network (NAMN) for sites with complete data runs of both NH₄⁺ and NH₃ over the specified time periods.**

| | NH₄⁺ (23 sites) (1999-2014) | NH₃ (23 sites) (1999-2014) | NH₄⁺ (30 sites) (2006-2014) | NH₃ (30 sites) (2006-2014) |
|---|---|---|---|---|
| UK emissions: % change over the time period | -16 (NH₃), -75 (SO₂), -53 (NO$_x$) | | -7 (NH₃), -54 (SO₂), -39 (NO$_x$) | |
| UK NAMN: % relative median change estimated from MK Sen's slope and intercept | -47** | 3.0 | -44** | -17 |
| UK NAMN: % relative change estimated from linear regression slope and intercept | -49** | 3.0 | -43** | -18^Δ |

Significance: *$p < 0.05$, **$p < 0.01$, ***$p < 0.001$, ^Δ$p = 0.06$.

**Table 7: Comparison of % change in UK NH₃, SO₂ and NO$_x$ emissions reported by the National Atmospheric Emission Inventory (NAEI) (data from*: http://naei.defra.gov.uk/*) with % change in annually averaged NH₄⁺ concentration data from the UK National Ammonia Monitoring Network (NAMN) and SO₄²⁻ and NO₃⁻ concentration data from the UK Acid Gas and Aerosol Network (AGANet) for sites with complete concurrent data runs over the specified time periods.**

| | NH₄⁺ (12 sites) (2000-2014) | SO4²⁻ (12 sites) (2000-2014) | NO₃⁻ (12 sites) (2000-2014) | NH₄⁺ (30 sites) (2006-2014) | SO4²⁻ (30 sites) (2006-2014) | NO₃⁻ (30 sites) (2006-2014) |
|---|---|---|---|---|---|---|
| UK emissions: % change over the time period | -16 (NH₃) | -75 (SO₂) | -53 (NO$_x$) | -7 (NH₃) | -54 (SO₂) | -39 (NO$_x$) |
| UK NAMN: % relative median change estimated from MK Sen's slope and intercept | -56** | -63*** | -46*** | -44** | -45* | -35** |
| UK NAMN: % relative change estimated from linear regression slope and intercept | -58** | -65*** | -45** | -43** | -46** | -33*** |

20  Significance: *$p < 0.05$, **$p < 0.01$, ***$p < 0.001$.