# Peer review of "Drivers for spatial, temporal and long-term trends in atmospheric ammonia and ammonium in the UK"

_Atmospheric Chemistry and Physics, 2017_

## Referee Comment (RC1) · L. Horváth (Referee) · 8 May 2017

MS is a good presentation of decrease of the ammonia emission in UK and the subsequent result of that measure. I believe it fits to the scope of ACP and merits to publish after revision. I suggest the following changes and improvements.

1) General observations

a) Emission sources According to the MS (Fig. 10b) the share of fertilizers is 16.2% together with "other animals". This is a simplification, these two sources have to split, since it takes 4-5 times higher contribution than that of sheep. So I miss displaying fertilizers from some figures (7, 10, 16).

b) Long-term trend analysis Ammonia emission in UK decreased substantially during the examined period while concentration remained at same level as it have been observed in other countries in Europe. Authors mentioned, it is the effect of sulphur dioxide emission and concentration decrease. It is true, but I miss a more detailed explanation of this mechanism. Fowler et al., 2001 (Water Air Soil Poll, Focus 1, 39-48) pointed out firstly the importance of co-deposition of ammonia and sulphur dioxide. I.e. there is a direct proportion between the SO2 concentration and the dry deposition velocity of NH3 onto natural surfaces that strongly influences the ammonia level in the atmosphere.

c) Seasonal trend analysis Source strength of ammonia – of course – strongly depends on temperature, so seasonal trend of NHx is mainly determined by this factor. But, as to the ammonia/ammonium transformation it is partly an equilibrium process due to the NH3 + HNO3 ↔ NH4NO3 reaction as it mentioned in the first paragraph of 3.5.6. The dissociation constant of ammonium nitrate depends on temperature, relative humidity and particle size (Mozurkewich, Atmos Envir 27A:261-270, 1993). At low relative air humidity (r.h. <60-70) ammonium nitrate does not exist in air at all. This phenomenon may strongly effect on seasonal variation of NH3 and NH4+ concentrations as a consequence of difference of summer/winter humidity even if part of ammonium is associated with hydrogen sulphate or sulphate ions. Authors should also describe this mechanism in the interpretation of NHX seasonal trend. A sulphate/nitrate ratio in aerosol phase in different seasons would give a good qualitative picture. In Fig. 18a we can observe nitrate dominance against the sulphate (≈2:1 in case of ammonium nitrate-ammonium hydrogen sulphate regime) that underlines the importance of ammonium nitrate in controlling the ammonium/ammonia ratio. Spring maxima for particle ammonium has observed and explained by the effect of non-domestic (continental) sources (after Vieno et al., 2014). But, the reason of that did not mentioned. How is the possible mechanism responsible for high continental ammonium (or ammonia) concentrations and transport from the continent in spring?

[Figure]

d) Sampling networking Because the short lifetime of ammonia it is difficult to find a "representative" measurement site for comparison with modelled concentrations on a 5×5 km grid. Authors mention a reason of the discrepancy between modelled concentration for the whole UK and concentration for the grids involving one or more measurement sites. This happens in the low and high concentrations regimes (<0.5 and >3.0 μg/m3) in different directions (over- or underestimation), as it also appears clearly in Figure 5. Authors describe some reasons of that (page 9 lines 39-41 though page 10 lines 1-2), mentioning that samplings were influenced by nearby emission sources. In this case some sites are not representative for the given 5×5 km grid. It is illustrated by the relationship between modelled and measured concentrations in the lower range (selected for the range of between the range of 0 and 4.5 μg/m3) where the relationship is stronger. Further analysis is needed how FRAME model correlated with measured NH3 concentrations in the work of Dore et al. 2015. Is there any discrepancy between modelled and measured concentrations in low and high ranges? How the model was validated? At sites with low concentrations samplings were performed in a clearing of forests. Question is: do model predict concentrations for layer above the canopy or for the ground level, where effect of deposition of the nearby forest is substantial? It would be the source of another bias between the modelled and measured ammonia concentrations. Other possible source of bias could be derived by the difference between the monthly sampling applied in the NAMN network and the sampling/measurement method for the validation of model. Are there any inter-comparison among the methods described in this manuscript and other methods based on daily or shorter time basis? In any case, taking into account that the modelled and measured concentrations agree well in middle range and in the average for the whole UK, the network seems to be suitable to establish trends for ammonia/ammonium concentrations.

e) Methods The sampling and analytical methods need more detailed descriptions. Detection limit precision, sensitivity if any should be mentioned for all sampling and analytical procedures (where appropriate).

f) Interpretations Manuscript has too many figures and tables. I suggest to reduce them. For example Fig 11 relationships among rainfall amount, temperature and ammonia emission can hardly be seen. Moreover this kind of relations have still demonstrated by Fig. 9. Also, for figures 12 and 13. One of them is unnecessary. It should be decided what is the more representative interpretation statistically, the trend of yearly or monthly data. I believe the latter. Do not repeat information both by figures and in tables. Statistical parameters are displayed in figures and also in tables (e.g. figures 13, 14, 15 and corresponding tables). Also there are redundancies with figures 17a and 18a.

g) Conclusion Too long and overlaps with discussions. It has to cut insisting only on the most important findings.

2) Specific comments:

Page 2:

First paragraph: Authors should describe the mechanism, how SO2 reduction influences the concentration and deposition of ammonia; here or/and in line 31, in 3.5.6., line 15 on page 17. Namely the decreased efficiency of co-deposition of SO2 and NH3 onto surfaces.

Line 13: SO2 and NOX are not acids, but acid anhydrides (as to SO2 and NO2).

Line 13: emitted "mainly" form combustion processes. (Do not forget natural sources esp. on global scale).

Line 14: Primary product of neutralization is the NH4HSO4 followed by forming (NH4)2SO4 only in case when ammonia is available in quantity enough.

Line 16: do not forget the role of PM in cloud/for formation as condensation nuclei.

Page 4:

Lines 17-18: The sentence "the network has a good representation in the middle air

concentration classes of 3-4 $\mu$g m-3" does not agree with Fig. 1c where measured concentration in the range of 3-4.5 $\mu$g m-3 is doubled. I would state instead "the network has a good representation in the middle air concentration classes of 0.5-3 $\mu$g m-3", so it is true. Otherwise it would make questionable the statement in lines 24-25, but this correlation should be justified also by figures.

Page 5: line 18: clarify the filter pack. I suppose the first filter is a Teflon one to capture particles.

Page 7: lines 12-13: was the two instruments inter-calibrated?

Page 18, lines 4-5: the formation of ammonia takes place by the same procedure with the same kinetic parameters, so cannot be "slower" rather less effective.

Fig. 4 is not demonstrative to me, e.g. the relation between discrete measurement points and emission map is hardly seen. An iso-line picture for ammonia instead of discrete figures would show better the situation, but the comparison of emission with concentration in this way has not much sense, because the effect of transport and transformation processes.

Figures involving temperature relating ammonia concentration: not mentioned but I believe they are air temperatures. But, emission of ammonia rather depends on soil surface temperature since decomposition of manure happens in the upper layer of soil. I know, soil temperature strongly correlates with air temperature but it has to be mentioned.

Figure 16b: mean NH3 of what? Square bracket suggests it is molar concentration, but mass concentration was used all over the MS. Better to name "NH3 concentration" on the axis and avoid bracket. How concentrations in Fig. 16b were calculated? Did authors split the ammonia concentration among the number of animals, taking into account the variation of the latter? How other sources were taken into account? Others than cows, pigs, poultry takes 1/3 of total emission. What does it mean "Total" in Fig.
Interactive comment

16. I suppose this is the total of cows, pigs, and poultry only rather than total emission from all of sources. Otherwise the blue line on Fig 16b should be uniform with pink one on 17a. Explain please in the legend. On the other hand concentrations does not directly relate to emission to compare.

Fig. 18a: nanomoles per what? Cubic meter?

3) Technical comments

Figures: use unambiguous and uniform in legends of vertical axes. E.g. concentration or emission of something (dimension in bracket).

Fig. 7, 8: Split the two figures (a and b), vertical axis of "b" is too close to "a"

Use greek mü instead u for micro in all figures

---

## Referee Comment (RC2) · Anonymous Referee #2 · 9 Jun 2017

The trend analysis in this study is superficial and does not meet a criteria for publishing in a high-impacted journals such as ACP. The authors are strongly encouraged to conduct a literature review for which trend analysis tools are the most suitable for this work. Incorrect adopting trend analysis tools also leads that several discussion such as "Trends in NH3 concentrations vs trends in NH3 emissions", "Influence of climate" and "Influence of local emission sources" is full of augments and lack of solid scientific values. The reviewer believes a substantial revision to be required to make the current version publishable.

[Figure]

2017.

---

## Author Comment (AC2) · 28 Aug 2017

RESPONSE TO REVIEWER

We have carefully considered Referee #2's comments.

Reviewer Comment: "The trend analysis in this study is superficial and does not meet a criteria for publishing in a high-impacted journals such as ACP."

Author response: The objective of the statistical trend analysis presented in our research paper was to identify trends in the long-term datasets (univariate monotonic, see e.g. Hirsch et al., 1991), estimate the rate of change and to address

the question of whether trends in NH3 and NH4+ concentrations (if any) are consistent with the changes in estimated UK annual NH3 emissions (data downloaded from: http://naei.beis.gov.uk/data/data-selector-results?q=101505)?" The dataset is sufficiently long-term (i.e. gaseous NH3: 17 years and particulate NH4+: 16 years) and collected by consistent methods, to allow for effective statistical trend analyses to be carried out.

To identify and quantify monotonic trends in the paper, trend assessment was carried out using (i) linear regression (LR), (ii) Mann-Kendall (MK) test (Hirsch et al., 1981; Gilbert, 1987) on annually averaged and monthly mean data, and (iii) Seasonal Mann-Kendall (SMK) test (Hirsch et al., 1982) on monthly data only. We think that this is not a superficial trend analysis - rather we applied the relevant methodologies. We referred to overviews of some of the more widely used techniques in time series modelling and analysis are widely available (see e.g., Chatfield, 2016; Hamilton, 1994; Meals et al., 2011). Online resources (e.g. https://cran.r-project.org/web/views/TimeSeries.html) also provide information on the range of statistical tests to identify and quantify trends in environmental data. It is noted that the non-parametric Mann-Kendall (MK) statistical approach is also commonly employed to detect monotonic trends in series of environmental data in many papers and scientific reports (e.g. Colette et al., 2012., Gurreiro et al., 2014, Li et al., 2016, Meals et al., 2011; Serrano et al., 1999; Torseth et al., 2012., Yao et al., 2016) and hydrological data (e.g. Hirsch et al., 1981, 1982). Trend analysis using the Mann-Kendall approach are also described in publications by ACP (e.g., Gurreiro et al., 2014, Li et al., 2016, Torseth et al., 2012., Yao et al., 2016). The advantages of the MK approach over linear regression for trend assessments are in that (i) it does not require normally distributed data, (ii) it is not affected by outliers, and (iii) it removes the effect of temporal auto-correlation in the data. The Seasonal Kendall test deployed also is highly robust and relatively powerful, recommended for water quality trend monitoring (Meals et al., 2011) and most recently applied in air pollution trend assessments in Europe (Colette et al., 2016; Torseth et al., 2012).

The causes of observed trends were subsequently interpreted in terms of three main drivers: 1) Meteorological: influence of temperature/rainfall 2) Changes in emissions from 3 dominant source sectors (cattle, pigs & poultry, sheep) 3) Changes in chemical climate, e.g. effects of large decrease in SO2 emissions and concentrations on co-deposition relationship of NH3 with SO2, and shift in form of particulate NH4+ from (NH4)2SO4 to NH4NO3.

It is noted that the MK and linear trend approach have been used in EMEP and in UK Air quality monitoring network reports respectively, therefore it was important to look at both and the differences.

Reviewer comment: "The authors are strongly encouraged to conduct a literature review for which trend analysis tools are the most suitable for this work."

Author Response: A literature review for trend analysis tools as suggested by the reviewer is considered outwith the scope of this research paper. We have added a sentences discussing the previous use of trend analysis methods by TFMM/EMEP and in UK Air quality monitoring network reports - primary users of these datasets. As noted in the text of the manuscript both analysis methods lead to similar results.

Reviewer Comment: "Incorrect adopting trend analysis tools also leads that several discussion such as "Trends in NH3 concentrations vs trends in NH3 emissions", "Influence of climate" and "Influence of local emission sources" is full of augments and lack of solid scientific values. The reviewer believes a substantial revision to be required to make the current version publishable"

Author response: Given our opinion that we have used appropriate methods, and a lack of detailed critique by Reviewer #2, we are unable to directly respond to this comment. However we do not think a substantial revision of our manuscript is required.

References

Chatfield, C.: The analysis of time series: an introduction. CRC press, 2016.

Colette, A., Aas, W., Banin, L., Braban, C.F., Ferm, M., Ortiz, A., Ilyin, I., Mar, K., Pandolfi, M., Putaud, J.P., and Shatalov, V.: Air pollution trends in the EMEP region between 1990 and 2012. Joint Report of the EMEP Task Force on Measurements and Modelling (TFMM), Chemical Co-ordinating Centre (CCC), Meteorological Synthesizing Centre-East (MSC-E), Meteorological Synthesizing Centre-West (MSC-W), EMEP/CCC-Report 1/2016, 2016.

Gilbert, R. O.: Statistical methods for environmental pollution monitoring. New York, John Wiley & Sons. 1987.

Guerreiro, C. B. B., Foltescu, V., and de Leeuw F.: Air quality status and trends in Europe, Atmospheric Environment., 98, 376-384, http://dx.doi.org/10.1016/j.atmosenv.2014.09.017, 2014.

Hamilton, J. D.: Time series analysis. Vol. 2. Princeton: Princeton university press, 1994.

Hirsch, R.M., Alexander, R.B., and Smith, R.A.: Selection of methods for the detection and estimation of trends in water quality. Water resources research, 27(5), 803-813, 1991.

Hirsch, R.M., Slack, J.R. and Smith, R.A.: Techniques of trend analysis for monthly water quality data. Water resources research, 18(1), 107-121, 1982.

Li, C., Martin, R. V., Boys, B. L., van Donkelaar, A., and Ruzzante, S.: Evaluation and application of multi-decadal visibility data for trend analysis of atmospheric haze, Atmos. Chem. Phys., 16, 2435-2457, https://doi.org/10.5194/acp-16-2435-2016, 2016

Meals, D.W., Spooner, J., Dressing, S.A., and Harcum, J.B., Statistical analysis for monotonic trends, Tech Notes 6, November 2011. Developed for U.S. Environmental Protection Agency by Tetra Tech, Inc., Fairfax, VA, 23 p, 2011. Available online at https://www.epa.gov/sites/production/files/2016-05/documents/tech_notes_6_dec2013_trend.pdf.

Serrano, A., Mateos, V.L.and Garcia, J.A.: Trend analysis of monthly precipitation over the iberian peninsula for the period 1921–1995, Physics and Chemistry of the Earth, Part B: Hydrology, Oceans and Atmosphere, 24(1), 85-90, 10.1016/S1464-1909(98)00016-1, 1999.

Tørseth, K., Aas, W., Breivik, K., Fjæraa, A. M., Fiebig, M., Hjellbrekke, A. G., Lund Myhre, C., Solberg, S., and Yttri, K. E.: Introduction to the European Monitoring and Evaluation Programme (EMEP) and observed atmospheric composition change during 1972-2009. Atmospheric Chemistry and Physics, 12, 5447-5481, http://dx.doi.org/10.5194/acp-12-5447-2012, 2012

Yao, X. and Zhang, L.: Trends in atmospheric ammonia at urban, rural, and remote sites across North America, Atmos. Chem. Phys., 16, 11465-11475, https://doi.org/10.5194/acp-16-11465-2016, 2016.

---

## Author Response (AR1)

**RESPONSE TO REVIEWERS**

**REVIEWER 1:**

**L. Horváth (Referee)**

The authors thank Prof. Horváth for his constructive comments and for taking the time to look at all the details described in the manuscript. We have carefully considered all comments. Please refer to the specific responses.

*1) General observations*
*"Emission sources According to the MS (Fig. 10b) the share of fertilizers is 16.2% together with 'other animals'.*
10 *This is a simplification, these two sources have to split, since it takes 4-5 times higher contribution than that of sheep. So I miss displaying fertilizers from some figures (7, 10, 16)."*

**Author Response:** There is one NAMN site only classed as dominated by emissions from the "fertiliser" emission source sector (see section 3.1 describing the classification of each NAMN sites according to one of seven specific
15 dominant emission source sectors). At this site, $NH_3$ is measured but not $NH_4^+$ (see Supp. Fig. 2a). Temporal and trend analysis have therefore not been carried out for the "fertilisers" source sector. The focus of Fig. 10b is on three specific dominant emission source sectors (Cattle, Pigs & Poultry, Sheep), compared with other emissions.

Since the "fertiliser" category is not considered in the paper (due to too few sites), "other animals + fertilisers" in
20 the pi-chart have been grouped together for simplicity. "Fertiliser" category has not been considered separately in Figs 7 and 16 for the same reason.

UK annual ammonia emissions data are downloaded from http://naei.defra.gov.uk/data/:
Other animals (horses) = 1.39 % and fertilisers = 14.86 %.
25 Sum of "other animals + fertiliser" = 16.2 % (Fig. 10b).

*"Long-term trend analysis Ammonia emission in UK decreased substantially during the examined period while*
30 *concentration remained at same level as it have been observed in other countries in Europe. Authors mentioned, it is the effect of sulphur dioxide emission and concentration decrease. It is true, but I miss a more detailed explanation of this mechanism. Fowler et al., 2001 (Water Air Soil Poll, Focus 1, 39-48) pointed out firstly the importance of co-deposition of ammonia and sulphur dioxide. I.e. there is a direct proportion between the $SO_2$ concentration and the dry deposition velocity of NH3 onto natural surfaces that strongly influences the ammonia*
35 *level in the atmosphere.*

**Author Response:** See response to Reviewer Comment 2 (p9).

*"Seasonal trend analysis Source strength of ammonia – of course – strongly depends on temperature, so seasonal trend of NHx is mainly determined by this factor. But, as to the ammonia/ammonium transformation it is partly an equilibrium process due to the NH3 + HNO3 : NH4NO3 reaction as it mentioned in the first paragraph of 3.5.6. The dissociation constant of ammonium nitrate depends on temperature, relative humidity and particle size (Mozurkewich, Atmos Envir 27A:261-270, 1993). At low relative air humidity (r.h. <60-70) ammonium nitrate does not exist in air at all. This phenomenon may strongly effect on seasonal variation of NH3 and NH4+ concentrations as a consequence of difference of summer/winter humidity even if part of ammonium is associated with hydrogen sulphate or sulphate ions. Authors should also describe this mechanism in the interpretation of NHX seasonal trend. A sulphate/nitrate ratio in aerosol phase in different seasons would give a good qualitative picture. In Fig. 18a we can observe nitrate dominance against the sulphate (_2:1 in case of ammonium nitrate-ammonium hydrogen sulphate regime) that underlines the importance of ammonium nitrate in controlling the ammonium/ammonia ratio. Spring maxima for particle ammonium has observed and explained by the effect of non-domestic (continental) sources (after Vieno et al., 2014). But, the reason of that did not mentioned. How is the possible mechanism responsible for high continental ammonium (or ammonia) concentrations and transport from the continent in spring?"*

**Author Response:** Section 3.3, paragraph 5 has been expanded to include explanation of the equilibrium between gas and aqueous aerosol phase as drivers in the seasonal variations of particulate $NH_4^+$.

"For particulate $NH_4^+$, as expected for a secondary pollutant, concentrations are more decoupled from the dominant $NH_3$ source sectors in the vicinity of a site. Although the formation of particulate $NH_4^+$ primarily depends on the occurrence of $NH_3$ in the atmosphere, synoptic meteorology and long range transboundary transport from continental Europe are important drivers influencing the seasonal variations of $NH_4^+$ across the UK, due to its' longer lifetime. The seasonal trends in particulate $NH_4^+$ are seen to be broadly similar for the four different emission source sectors (Figure 8b), with the magnitude of the $NH_4^+$ concentrations reflecting $NH_3$ concentrations at a regional level. In the atmosphere, particulate $NH_4^+$ are primarily in the form of $(NH_4)_2SO_4$ and $NH_4NO_3$, formed when the acid gases $HNO_3$ and $H_2SO_4$.in the atmosphere are neutralised by $NH_3$ (Putaud et al., 2010). $NH_3$ preferentially neutralizes $H_2SO_4$ due to its low saturation vapour pressure (forming $NH_4HSO_4$ followed by $(NH_4)_2SO_4$)), while $NH_4NO_3$ is formed when abundant $NH_3$ is available, In contrast to $(NH_4)_2SO_4$, $NH_4NO_3$ is a semi-volatile component (Stelson & Seinfeld, 1982). Long-term data from the UK Acid Gas and Aerosol Network (AGANet, Conolly et al., 2016) shows a change in the particulate phase of $NH_4^+$ from $(NH_4)_2SO_4$ to $NH_4NO_3$, with particulate nitrate concentrations exceeding that of particulate sulphate approximately three-fold (on a molar basis) (Fig. 18a). This suggests that the thermodynamic equilibrium between the gas phase $NH_3$ and $HNO_3$ and the aerosol phase $NH_4NO_3$ will have a much greater effect on the seasonal concentrations of $NH_4^+$ than $(NH_4)_2SO_4$. The formation and dissociation of $NH_4NO_3$ depend strongly on ambient temperature and humidity (Stelson and Seinfeld, 1982). Warm, dry weather in summer promotes dissociation, decreasing particulate phase $NH_4NO_3$ relative to gas phase $NH_3$ and $HNO_3$. During the winter months, low temperature and high humidity favour the formation of $NH_4NO_3$ from the gas phase $NH_3$ and $HNO_3$. By contrast, the spring peak in $NH_4^+$ concentrations may be attributed to photochemical processes (elevated ozone) leading to enhanced formation of $HNO_3$ during this period (Pope et al., 2016) and also to import of particulate $NO_3^-$ through long-range transboundary transport, e.g. from continental Europe, as discussed in Vieno et al. (2014). Nevertheless, it is notable that the winter minima for $NH_4^+$ aerosol concentrations at sheep and background sites are more pronounced than that for pig, poultry and cattle dominated sites. This may be a result of a combination of smaller $NH_3$ emissions in winter in these areas (as indicated by Figure 8a) and differences in long-range transport to the more remote areas in winter conditions."

References added:

Pope, R.J., Butt, E.W., Chipperfield, M.P., Doherty, R.M., Fenech, S., Schmidt, A., Arnold, S.R and Savage, N.H. The impact of synoptic weather on UK surface ozone and implications for premature mortality. Environmental Research Letters. 11, doi:10.1088/1748-9326/11/12/124004, 2016.

Putaud, J.P., Van Dingenen, R., Alastuey, A., Bauer, H., Birmili, W., Cyrys, J., Flentje, H., Fuzzi, S., Gehrig, R., Hansson, H.C. and Harrison, R.M. A European aerosol phenomenology–3: Physical and chemical characteristics of particulate matter from 60 rural, urban, and kerbside sites across Europe. *Atmospheric Environment*, *44*(10), 1308-1320, doi:10.1016/j.atmosenv.2009.12.011, 2010.

Stelson, A. W., and Seinfeld, J. H.: Relative humidity and temperature dependence of the ammonium nitrate dissociation constant, Atmospheric Environment, 16, 983-992, doi: 10.1016/0004-6981(82)90184-6, 1982.

*"Sampling networking. Because the short lifetime of ammonia it is difficult to find a "representative" measurement site for comparison with modelled concentrations on a 5_5 km grid. Authors mention a reason of the discrepancy between modelled concentration for the whole UK and concentration for the grids involving one or more measurement sites. This happens in the low and high concentrations regimes (<0.5 and >3.0 _g/m3) in different directions (over- or underestimation), as it also appears clearly in Figure 5. Authors describe some reasons of that (page 9 lines 39-41 though page 10 lines 1-2), mentioning that samplings were influenced by nearby emission sources. In this case some sites are not representative for the given 5_5 km grid. It is illustrated by the relationship between modelled and measured concentrations in the lower range (selected for the range of between the range of 0 and 4.5 _g/m3) where the relationship is stronger. __Further analysis is needed how FRAME model correlated with measured NH3 concentrations in the work of Dore et al. 2015. Is there any discrepancy between modelled and measured concentrations in low and high ranges? How the model was validated?__ At sites with low concentrations samplings were performed in a clearing of forests. Question is: __do model predict concentrations for layer above the canopy or for the ground level, where effect of deposition of the nearby forest is substantial?__ It would be the source of another bias between the modelled and measured ammonia concentrations. __Other possible source of bias could be derived by the difference between the monthly sampling applied in the NAMN network and the sampling/measurement method for the validation of model. Are there any inter-comparison among the methods described in this manuscript and other methods based on daily or shorter time basis?__ In any case, taking into account that the modelled and measured concentrations agree well in middle range and in the average for the whole UK, the network seems to be suitable to establish trends for ammonia/ammonium concentrations."*

*Re: "Further analysis is needed how FRAME model correlated with measured NH3 concentrations in the work of Dore et al. 2015. Is there any discrepancy between modelled and measured concentrations in low and high ranges? How the model was validated?"*

**Author Response:** Validation of the atmospheric transport model FRAME (Fine Resolution AMmonia Exchange) in estimating atmospheric concentrations and deposition rates of gaseous $NH_3$ and particulate $NH_4^+$ have previously been made by comparison with measurements from the UK NAMN (Dore et al., 2007) and by comparison with other models (Dore et al., 2015). When compared with other atmospheric chemistry transport models, FRAME was found to correlate well with measured $NH_3$ and $NH_4^+$ concentrations from NAMN (Dore et al., 2015). The comparison of NAMN $NH_3$ and $NH_4^+$ measurements with modelled $NH_3$ concentrations from the FRAME model in the paper is made for an example year of 2012 in the paper, This updates an earlier inter-comparison assessment carried out by Dore et al. (2007) for the year 2002, and demonstrates that the FRAME model is performing well in describing the spatial distribution of both $NH_3$ and $NH_4^+$.

The FRAME model uses a database of $NH_3$ emissions with a 5×5 km grid-square resolution as input. In the present comparison of the FRAME model estimates (based on 2012 UK AENEID $NH_3$ emission data) with the NAMN measurement results for 2012 (Figure 5), the network annual mean concentrations for each site is compared against the model estimate for the 5-km grid square in which it occurs, and the point is classified according to the estimated dominant source sector of the grid square. Both the model outputs and the measurement agree that background and sheep sites are characterised by small $NH_3$ concentrations (< 1 µg $NH_3$ m$^{-3}$ annual mean), while agricultural areas, particularly areas with intensive pig and poultry areas, are associated with large $NH_3$ concentrations (up to 8 µg $NH_3$ m$^{-3}$ annual mean). Overall, the comparison suggests a fairly good fit with regard to both the magnitude and spatial variability of $NH_3$ concentrations at a national scale, with an $R^2$ value of 0.6 (**Error! Reference source not found.**). The results of the network thus broadly support the predictions of the FRAME model, lending support to the AENEID model outputs. There are however, systematic differences in the comparison of FRAME and the measurements, depending on the air concentration and dominant source. Figure 5 shows that concentrations are overestimated by FRAME in areas dominated by cattle, pig and poultry, compared with the measurement data,

while the results agree well in sheep and non-agricultural areas. Possible reasons for the overestimation of cattle, pig and poultry farming by FRAME compared to the measurements may be due to the following:

a) spatial location of the sampling site relative to the distribution of sources. Ammonia exhibits large sub-grid spatial variability (Dragosits et al. 2002), with the result that single site measurements may not reflect the concentrations across the 5 km grid squares. For example, at many of the sites where the model overestimates concentrations, the measurements are in fact made in nature reserves, which would on average be more distant from sources than assumed in the FRAME 5 km average estimates, thereby underestimating concentrations, This effect is particularly important in areas with high local variability in ammonia emissions, such as intensive agricultural areas. This illustrates the importance of having a large number of sites for comparison,

b) accuracy of the emissions data that are critical to the performance of the model. For example accuracy of emission factors for different livestock classes affecting the model estimates, or

c) that dispersion in the model is slightly underestimated. Clearly further work is required to address these questions.

Dore, A. J., Vieno, M., Tang, Y. S., Dragosits, U., Dosio, A., Weston, K. J., & Sutton, M. A. (2007). Modelling the atmospheric transport and deposition of sulphur and nitrogen over the United Kingdom and assessment of the influence of SO 2 emissions from international shipping. *Atmospheric Environment*, *41*(11), 2355-2367.

Dore, A. J., Carslaw, D. C., Braban, C., Cain, M., Chemel, C., Conolly, C. & Lawrence, S. (2015). Evaluation of the performance of different atmospheric chemical transport models and inter-comparison of nitrogen and sulphur deposition estimates for the UK. *Atmospheric Environment*, *119*, 131-143.

*Re: "At sites with low concentrations samplings were performed in a clearing of forests. Question is: do model predict concentrations for layer above the canopy or for the ground level, where effect of deposition of the nearby forest is substantial? It would be the source of another bias between the modelled and measured ammonia concentrations."*

**Author Response:** FRAME is a Lagrangian model that incorporates horizontal and vertical gradients of $NH_3$ and calculates vertical concentration profiles with diffusion through 33 layers of varying depth. The modelled concentrations output is from the 1-2 m layer, used to compare with NAMN measurements that are made at approx. 1.5 m above ground.

Additional text has been added to Section 2.1, paragraph 1, after the first sentence to provide further information on siting of sites to provide representative measurements:

*"The network covers a wide distribution of monitoring sites with measurements in both agricultural and semi-natural areas. Monitoring locations are sited away from point sources (> 150 m) such as farm buildings, which avoids overestimating $NH_3$ concentrations compared with the grid square, since the aim is to provide meso-scale and regional patterns. In addition, where sampling is carried out in woodland areas, it is made in clearings."*

*Re: "Other possible source of bias could be derived by the difference between the monthly sampling applied in the NAMN network and the sampling/measurement method for the validation of model. Are there any inter-comparison among the methods described in this manuscript and other methods based on daily or shorter time basis?"*

**Author Response:** Comparison between model and measurement discussed in this paper is based on annual concentrations. Annual mean concentrations from NAMN are derived from the mean of monthly measured concentrations.

Inter-comparison among the methods described in this manuscript;

ALPHA vs Daily Annular Denuder Method – see section 2.2.2, last paragraph:
In the USA (Puchalski et al., 2011), the ALPHA samplers performed well against a reference annular denuder method with a median relative percent difference of −2.4%.

The ALPHA and DELTA methods described in this manuscript have also been compared with other methods with shorter time resolution and performed well. Some examples and references are given below.

Comparison with different methods with daily timescales:
DELTA vs Daily Annular Denuder method – see Tang et al., 2009
Tang, Y. S., Simmons, I., van Dijk, N., Di Marco, C., Nemitz, E., Dammgen, U., Gilke, K., Djuricic, V., Vidic, S., Gliha, Z., Borovecki, D., Mitosinkova, M., Hanssen, J. E., Uggerud, T. H., Sanz, M. J., Sanz, P., Chorda, J. V., Flechard, C. R., Fauvel, Y., Ferm, M., Perrino, C., and Sutton, M. A.: European scale application of atmospheric reactive nitrogen measurements in a low-cost approach to infer dry deposition fluxes, Agriculture Ecosystems & Environment, 133, 183-195, doi.org/10.1016/j.agee.2009.04.027, 2009.

DELTA vs Daily Filter Pack (EMEP method) – see Tang et al., 2017 (unpublished data, paper in prep)

[Figure]

Comparison of total inorganic ammonium, TIA (sum of $NH_3 + NH_4^+$) concentrations at the Eskdalemuir monitoring station (EMEP station code = GB0002R; UK-AIR ID = UKA00130) measured under the EMEP program with concentrations of the corresponding gas and aerosol from the UK National Ammonia Monitoring Network (NAMN, $NH_3$ and $NH_4^+$). EMEP values (data downloaded from http://ebas.nilu.no/) are means of daily measurements for TIA by the EMEP filter pack method, matched to the NAMN sampling periods (monthly). Filter pack measurements at Eskdalemuir terminated in December 2000.

Comparison with different methods with shorter timescales:
DELTA and ALPHA vs AMOR at Zegfeld (ID 633; Dutch National Air Quality Monitoring Network, LML):
An intercomparison of $NH_3$ measurements by the RIVM AMOR system (hourly) and the CEH DELTA sampling system (monthly) have been carried out at the Zegweld site in the Netherlands since July 2003. Since September 2012, CEH ALPHA measurements have also been included. To compare results, monthly mean concentrations were derived from the average of hourly AMOR data for the corresponding DELTA and ALPHA monthly sampling periods. The long-term comparison with the AMOR at Zegfeld, NL, has been added to the Supplementary Material. The following text is added to the Manuscript in Section 2.2.4 and the Graph is Supp. Figure S6)

"An intercomparison of $NH_3$ measurements by the RIVM AMOR system (hourly, Wyers et al., 1993;) and the DELTA sampling system (monthly) have been carried out at the Zegweld site (ID 633) in the Dutch National Air Quality Monitoring Network (van Zanten et al., 2017) since July 2003. Since September 2012, ALPHA measurements have also been included. To compare results, monthly mean concentrations were derived from the average of hourly AMOR data for the corresponding DELTA and ALPHA monthly sampling periods with good agreement (supp. Figure S6)."

*Re:* "*Methods The sampling and analytical methods need more detailed descriptions. Detection limit precision, sensitivity if any should be mentioned for all sampling and analytical procedures (where appropriate).*"

**Author Response:** Detailed descriptions of the DELTA and ALPHA methodologies are available from the references provided in the paper (Sutton et al., 2001a, 2001c and Tang et al., 2001a). We feel that sufficient analytical details have already been provided in section 2.2.1. DELTA methods and section 2,2.2. Passive Methods. Some additional text describing sample analysis and LODs has been added at end of Section 2.2.3,

Chemical analysis:

"The extracted samples were analysed for $NH_4^+$ against a series of $NH_4^+$ standards and quality controls. Parallel analysis of laboratory and field blank (unexposed) samples were used to determine the amounts of $NH_4^+$ derived from $NH_3$ and $NH_4^+$ in the atmosphere during transport and storage. The limit of detection (LOD) calculation of the ALPHA and DELTA methodologies are determined as three times the standard deviations of the laboratory blanks. For the DELTA method, the LODs were 0.01 $\mu g\ m^{-3}$ for gaseous $NH_3$ and 0.02 $\mu g\ m^{-3}$ for particulate $NH_4^+$. For the ALPHA method, the LOD was determined as 0.03 $\mu g\ m^{-3}$"

*Re:* "*Interpretations Manuscript has too many figures and tables. I suggest to reduce them. For example Fig 11 relationships among rainfall amount, temperature and ammonia emission can hardly be seen. Moreover this kind of relations have still demonstrated by Fig. 9. Also, for figures 12 and 13. One of them is unnecessary. It should be decided what is the more representative interpretation statistically, the trend of yearly or monthly data. I believe the latter. Do not repeat information both by figures and in tables. Statistical parameters are displayed in figures and also in tables (e.g. figures13, 14, 15 and corresponding tables). Also there are redundancies with figures 17a and 18a.*"

**Author Response:** The authors feel that there is justification for the number of graphs presented as each has a particular purpose. We have explained this for each graph below, though we have agreed that Figure 17a was redundant as similar data was also shown in Figure 18a.

*Re: "Figure 11"*

**Author Response:** Figure 1 demonstrates the relationships between monitored monthly mean $NH_3$ concentrations with locally available monthly temperature and rainfall data at an example site. Figure 11 on the other hand provides an important comparison on a national level the annual mean $NH_3$ concentrations of all NAMN sites with UK annual mean temperature and rainfall. We strongly suggest to retain this graph, since it shows the strong inter- and intra-annual variability in the parameters considered. The annually averaged data of all sites masks considerable spatial and seasonal variability in $NH_3$ concentrations. Drivers contributing to this variability include the influence of climate on emissions, variations in management practice for a particular emission source, and influence of local emission sources and chemical interactions with other chemical species on $NH_3$ and $NH_4^+$ concentrations that are discussed in the paper.

*Re: "Figures 12 and 13 - one of them is unnecessary. It should be decided what is the more representative interpretation statistically, the trend of yearly or monthly data. I believe the latter"*

**Author Response:** The authors feel that both annual and seasonal variability are of equal interest. Since ammonia is strongly seasonally variable, it is important to demonstrate this graphically. In addition, the monthly data allows the seasonality to be accounted for in the seasonal Mann-Kendall test, which takes into account the 12 month seasonality and tests whether there is a trend not due to seasonality.

*Re: "Statistical parameters are displayed in figures and also in tables (e.g. figures13, 14, 15 and corresponding tables)."*

**Author Response:** The data has been shown graphically and in table form in order for data transparency and so that readers may use the parameterisations if they wish.

*Re: "Also there are redundancies with figures 17a and 18a."*

**Author Response:** Agree, Figure 17a deleted.

*Re: "Conclusion Too long and overlaps with discussions. It has to cut insisting only on the most important findings."*

**Author Response:** We feel that the conclusions are not too long and are not particularly discursive. Therefore we would prefer to leave them as they are.

*"Page 2: First paragraph: Authors should describe the mechanism, how SO2 reduction influences the concentration and deposition of ammonia; here or/and in line 31, in 3.5.6.,line 15 on page 17. Namely the decreased efficiency of co-deposition of SO2 and NH3 onto surfaces."*

**Author Response**:  the following text has been added section 3.5.6, after paragraph 2:

*"Dry deposition of $SO_2$ and $NH_3$ are enhanced in the presence of both gases, an interaction referred to as "co-deposition" (Fowler et al., 2001). The acid-base neutralization by each of the gases provides an efficient sink for dry deposition on leaf surfaces and deposition enhancement for each gas depends on the relative air concentrations of $NH_3$ and $SO_2$. For $SO_2$, the dry deposition process has been shown to be strongly influenced by ambient concentrations of $NH_3$ because the surface resistance is regulated mainly by uptake in moisture on foliar surfaces, which, in turn, is strongly influenced by the presence of $NH_3$. The large reduction in $SO_2$ emissions and ambient concentrations, compared with the relative stagnation in $NH_3$ emissions and concentrations over the same period has meant that the $SO_2/NH_3$ ratio has decreased dramatically. This has led to a systematic decrease in canopy resistance to uptake of $SO_2$ on surfaces, increasing dry deposition of $SO_2$ in the UK (ROTAP 2012). The underlying cause of the decrease in surface resistance is that the ambient $NH_3$ is sufficient to neutralize acidity from the solution and oxidation of deposited $SO_2$, maintaining large rates of deposition."*

*"Line 13: SO2 and NOx are not acids, but acid anhydrides (as to SO2 and NO2)."*

**Author Response**: text changed from "acids" to "acid gases"

*"Line 13: emitted "mainly" form combustion processes. (Do not forget natural sources esp. on global scale)."*

**Author Response**: text added "and from natural sources"

*"Line 14: Primary product of neutralization is the NH4HSO4 followed by forming (NH4)2SO4 only in case when ammonia is available in quantity enough."*

**Author Response**: text added. See Author response 1c. on page 2.

*"Line 16: do not forget the role of PM in cloud/for formation as condensation nuclei."*

**Author Response**: We have modified the sentence ….
*"The effects of PM on atmospheric visibility, radiative scattering, cloud formation (and resultant climate effects)..."*

Page 4:
*"Lines 17-18: The sentence "the network has a good representation in the middle air concentration classes of 3-4 ug m-3" does not agree with Fig. 1c where measured concentration in the range of 3-4.5 ug m-3 is doubled. I would state instead "the network has a good representation in the middle air concentration classes of 0.5-3 ug m-3", so it is true. Otherwise it would make questionable the statement in lines 24-25, but this correlation should be justified also by figures."*

**Author Response:** We changed the sentence to:

> *"..the network has a good representation in the middle air concentration classes of 0.5 – 1.5 µg m$^{-3}$ (33 % of NAMN sites, compared with 29 % of all FRAME 5 km x 5 km grid squares) and 1.5 - 3 µg m$^{-3}$ (32 % of NAMN sites, compared with 39 % of all FRAME 5 km x 5 km grid squares), but with …. (Figure 2c).*

"Page 5: line 18: *clarify the filter pack. I suppose the first filter is a Teflon one to capture particles."*
**Author Response:** The filter is cellulose impregnated with citric acid. The word "cellulose" has been added to the sentence. There is no second filter.

*"Page 7: lines 12-13: was the two instruments inter-calibrated?"*
**Author Response:** Yes the passive diffusion tube and ALPHA method are calibrated against the active sampling DELTA method on a monthly basis as discussed in Section 2.2.2 paragraph 3.

*"Page 18, lines 4-5: the formation of ammonia takes place by the same procedure with the same kinetic parameters, so cannot be "slower" rather less effective."*
**Author Response:** The authors agree and think this was a typo. We have revised the test changing "slower" to "lower"

*"Fig. 4 is not demonstrative to me, e.g. the relation between discrete measurement points and emission map is hardly seen. An iso-line picture for ammonia instead of discrete figures would show better the situation, but the comparison of emission with concentration in this way has not much sense, because the effect of transport and transformation processes."*

*Re: "An iso-line picture for ammonia instead of discrete figures would show better the situation"*

**Author Response:** Interpolated concentration maps have not been used since the interpolation of the discrete measurement points (e.g. using bilinear interpolation) will give the reader a false sense of the spatial variability of air concentrations from the limited number of measurement locations. The number of sites required to replicate the spatial resolution of the emissions map (5 km x 5 km grid resolution) will be impossibly high. The measured annual concentrations have therefore been shown as coloured dots on the map to show the observed spatial variability across the UK. Interpolated concentration maps can be produced from the discrete measurement points and added in Supplementary Materials, if required.

*Re: "but the comparison of emission with concentration in this way has not much sense, because the effect of transport and transformation processes"*

**Author Response:** The authors acknowledge that concentrations are affected by effect of transport and transformation processes, but at the same time, concentrations are also largely driven by emissions. The large variability in $NH_3$ emissions across the UK is reflected by both modelled (FRAME) and observed spatial variability in $NH_3$ and $NH_4^+$ concentrations, with largest concentrations in the largest emission source areas and lowest in background areas. The comparison of the measured concentrations (shown as discrete point data) with the emissions maps support this in Figure 4.

The FRAME model uses as input annual ammonia emissions data from the UK National Atmospheric Emissions Inventory (http://naei.defra.gov.uk/) and incorporates the main atmospheric processes (emission, diffusion, chemistry and deposition) to calculate annual $NH_3$ concentration fields in the UK at a 5 km x 5 km grid resolution. The spatial variability estimated by the FRAME model mirrors the variability in emissions across the country, with largest concentrations in the largest emission source areas.

*"Figures involving temperature relating ammonia concentration: not mentioned but I believe they are air temperatures. But, emission of ammonia rather depends on soil surface temperature since decomposition of manure happens in the upper layer of soil. I know, soil temperature strongly correlates with air temperature but it has to be mentioned."*

**Author Response:** We confirm that the temperatures used are air temperatures, which are more available across the domain than soil temperatures. As Prof. Horvath points out, the soil and air temperatures do correlate.

*"Figure 16b: mean NH3 of what? Square bracket suggests it is molar concentration, but mass concentration was used all over the MS. Better to name "NH3 concentration" on the axis and avoid bracket."*

**Author Response:** The authors agree and have adjusted the graph axis label and the figure caption.

*"How concentrations in Fig. 16b were calculated? Did authors split the ammonia concentration among the number of animals, taking into account the variation of the latter? How other sources were taken into account? Others than cows, pigs, poultry takes 1/3 of total emission. What does it mean "Total" in Fig.16. I suppose this is the total of cows, pigs, and poultry only rather than total emission from all of sources. Otherwise the blue line on Fig 16b should be uniform with pink one on 17a. Explain please in the legend. On the other hand concentrations does not directly relate to emission to compare."*

**Author Response:** The authors apologise, we omitted a description of Figure 16 in the text. This has now been added to Section 3.5.5 paragraph 3 as follows:

"In Figure 16, the relative changes in UK emissions between 1998 and 2014 are compared with relative changes in mean measured $NH_3$ concentrations for all NAMN sites, and for grouped sites classified as dominated by cattle, pigs & poultry, and sheep."

*Fig. 18a: nanomoles per what? Cubic meter?*

**Author Response:** Nanomoles per cubic metre. Axis on graph corrected.
*3) Technical comments*
*"Figures: use unambiguous and uniform in legends of vertical axes. E.g. concentration or emission of something (dimension in bracket)."*

**Author Response:** Most graphs are systematically labelled but we have adjusted Figure 18 to put the percentage in brackets.

*"Fig. 7, 8: Split the two figures (a and b), vertical axis of "b" is too close to "a""*

**Author response:**  Thank you for spotting this. We have adjusted the a and b so they are separated more widely.

*"Use greek mü instead u for micro in all figures"*

**Author Response:** Yes, we have checked and updated all figures.
Figures 8a, 8b corrected
Supp. Figure 1a, 1b corrected

REVIEWER 2:

**Anonymous Referee #2**

**RESPONSE TO REVIEWER**

We have carefully considered Referee #2's comments.

*"The trend analysis in this study is superficial and does not meet a criteria for publishing in a high-impacted journals such as ACP."*

The objective of the statistical trend analysis presented in our research paper was to identify trends in the long-term datasets (univariate monotonic, see e.g. Hirsch et al., 1991), estimate the rate of change and to address the question of whether trends in $NH_3$ and $NH_4^+$ concentrations (if any) are consistent with the changes in estimated UK annual $NH_3$ emissions (data downloaded from: http://naei.beis.gov.uk/data/data-selector-results?q=101505)?" The dataset is sufficiently long-term (i.e. gaseous $NH_3$: 17 years and particulate $NH_4^+$: 16 years) and collected by consistent methods, to allow for effective statistical trend analyses to be carried out.

To identify and quantify monotonic trends in the paper, trend assessment was carried out using (i) linear regression (LR), (ii) Mann-Kendall (MK) test (Hirsch et al., 1981; Gilbert, 1987) on annually averaged and monthly mean data, and (iii) Seasonal Mann-Kendall (SMK) test (Hirsch et al., 1982) on monthly data only. We think that this is not a superficial trend analysis - rather we applied the relevant methodologies. We referred to overviews of some of the more widely used techniques in time series modelling and analysis are widely available (see e.g., Chatfield, 2016; Hamilton, 1994; Meals et al., 2011). Online resources (e.g. https://cran.r-project.org/web/views/TimeSeries.html) also provide information on the range of statistical tests to identify and quantify trends in environmental data.

It is noted that the non-parametric Mann-Kendall (MK) statistical approach is also commonly employed to detect monotonic trends in series of environmental data in many papers and scientific reports (e.g. Colette et al., 2012., Gurreiro et al., 2014, Li et al., 2016, Meals et al., 2011; Serrano et al., 1999; Torseth et al., 2012., Yao et al., 2016) and hydrological data (e.g. Hirsch et al., 1981, 1982).

Trend analysis using the Mann-Kendall approach are also described in publications by ACP (e.g., Gurreiro et al., 2014, Li et al., 2016, Torseth et al., 2012., Yao et al., 2016). The advantages of the MK approach over linear regression for trend assessments are in that (i) it does not require normally distributed data, (ii) it is not affected by outliers, and (iii) it removes the effect of temporal auto-correlation in the data. The Seasonal Kendall test deployed also is highly robust and relatively powerful, recommended for water quality trend monitoring (Meals et al., 2011) and most recently applied in air pollution trend assessments in Europe (Colette et al., 2016; Torseth et al., 2012).

The cause of the trends were subsequently interpreted in terms of three main drivers:
   1) Meteorological: influence of temperature/rainfall
   2) Changes in emissions from 3 dominant source sectors (cattle, pigs & poultry, sheep)
   3) Changes in chemical climate, e.g. effects of large decrease in $SO_2$ emissions and concentrations on co-deposition relationship of $NH_3$ with $SO_2$, and shift in form of particulate $NH_4^+$ from $(NH_4)_2SO_4$ to $NH_4NO_3$.

*"The authors are strongly encouraged to conduct a literature review for which trend analysis tools are the most suitable for this work."*

A literature review for trend analysis tools as suggested by the reviewer is considered outwith the scope of this research paper.

We have added a sentences discussing the previous use of trend analysis methods by EMEP and in UK Air quality monitoring network reports - primary users of these datasets. As noted in the text of the manuscript both analysis methods lead to similar results.

Incorrect adopting trend analysis tools also leads that several discussion such as "Trends in NH3 concentrations vs trends in NH$_3$ emissions", "Influence of climate" and "Influence of local emission sources" is full of augments and lack of solid scientific values. The reviewer believes a substantial revision to be required to make the current version publishable

Given our opinion that we have used appropriate methods, and a lack of detailed critique by Reviewer #2, we are unable to directly respond to this comment, however do not think a substantial revision is required.

**Manuscript changes:**

- Section 1 (paragraph 2): text changed from "acids" to "acid gases"
- Section 1 (paragraph 2): text added "and from natural sources"
- Section 1 paragrpah 2: Text modified:

[revised manuscript text omitted]

---

## Author Response (AR2)

**RESPONSE TO REVIEWER**

1) "The current version of the manuscript is lack of solid evidences and full of speculation and arguments.

**Author Response:** We have addressed comments raised by both reviewers and revised the manuscript accordingly. Reviewer 1 was satisfied with latest version of the manuscript. We have carefully considered further comments raised by reviewer 2 in the following pages. Please refer to the specific responses.

2) "What the authors selected the statistical analysis tool appears to be quietly at will, at least to this reviewer. The authors failed to demonstrate why the selected statistical analysis tool can allow addressing the scientific questions associated with their specific data sets."

**Author Response:** To reiterate, the long-term continuous monthly measurement data of atmospheric $NH_3$ and $NH_4^+$ from the UK National Ammonia Monitoring Network was used in the manuscript to establish whether, and the extent to which, concentrations of these air pollutants have changed in relation to the estimated decrease in $NH_3$ emissions over the time period where data is available. With a long data record, trend analysis represents the best approach to evaluate gradual change in air pollutant concentrations resulting from UK-wide emission changes with time. The two basic types of trends that can be statistically analyzed are monotonic and step trends (see e.g. Hirsch et al., 1991). Monotonic trends are generally gradual changes that are either increasing or decreasing with no reversal of direction. Step trends are where there is an abrupt shift at a specific point in time.

Monotonic trend assessment on annual mean data was conducted since:
1) UK emissions data are annual. Annual trends in air pollutant data was thus investigated in relation to annual trends in emissions data,
2) UK-wide changes in $NH_3$ emissions over time has been gradual with no abrupt changes. Plotting the long-term data show either none or gradual changes that are either increasing or decreasing with no reversal of direction.

Parametric (e.g. linear regression) and non-parametric (e.g. Mann-Kendall) tests are commonly used for monotonic trend assessment in environmental data.

See also our response (excerpt copied below) to your previous comment *"The trend analysis in this study is superficial and does not meet a criteria for publishing in a high-impacted journals such as ACP."*
"The objective of the statistical trend analysis presented in our research paper was to identify trends in the long-term datasets (univariate monotonic, see e.g. Hirsch et al., 1991), estimate the rate of change and to address the question of whether trends in $NH_3$ and $NH_4^+$ concentrations (if any) are consistent with the changes in estimated UK annual $NH_3$ emissions (data downloaded from: http://naei.beis.gov.uk/data/data-selector-results?q=101505)?" The dataset is sufficiently long-term (i.e. gaseous $NH_3$: 17 years and particulate $NH_4^+$: 16 years) and collected by consistent methods, to allow for effective statistical trend analyses to be carried out.

To address the concerns of reviewer 2 regarding the selection of statistical analysis tools, additional text (highlighted) has been added in <section 2.2.5. Trend Analyses> below:

Statistical trend analysis was conducted on the long-term dataset from the UK NAMN to identify trends in the long-term datasets (univariate monotonic, see e.g. Hirsch et al., 1991), estimate the rate of change and to address the question of whether trends in $NH_3$ and $NH_4^+$ concentrations (if any) are consistent with the changes in estimated UK annual $NH_3$ emissions (data downloaded from: http://naei.beis.gov.uk/data/data-selector-results?q=101505)? The dataset is sufficiently long-term (i.e. gaseous $NH_3$: 17 years and particulate $NH_4^+$: 16 years) and collected by consistent methods, to allow for effective statistical trend analyses to be carried out.

Trend analyses were carried out using (i) linear regression (LR), (ii) Mann-Kendall (MK) test (Gilbert, 1987) on annually averaged and monthly mean data, and (iii) Seasonal Mann-Kendall (SMK) test (Hirsch et al., 1982) on monthly data only. Mann-Kendall tests were performed using the 'Kendall' package (McLeod, 2015) in the R software. Computation of the Sen's slope and confidence interval (for non-seasonal Sen's slope only) of the linear trend were performed using the R 'Trend' package (Pohlert, 2016).

Since concentrations of $NH_3$ show strong seasonality, the SMK test was applied to identify the months that are driving the long-term trends in data. The SMK test (Hirsch et al., 1982) takes into account a 12 month seasonality in the time series data by computing the MK test on each of monthly 'seasons' separately, and then combining the results. So for monthly 'seasons', January data are compared only with January, February only with February, etc. No comparisons are made across season boundaries. The Sen's slope is the fitted median slope of a linear regression joining all pairs of observations. For the SMK, an estimate of the seasonal Sen's trend slope over time is computed as the median of all slopes between data pairs within the same season (i.e. January compared only with January etc.). Therefore no cross-season slopes contribute to the overall estimate of the SMK trend slope.

Parametric LR analysis are simple and straightforward to use and interpret monotonic trend assessment in environmental data (e.g. Kindzierski et al., 2009; Meals et al., 2011), but they require assumptions about normality of data and homogeneity of variance of data. The MK approach on the other hand are widely used in environmental time series assessments, e.g. long-term trends in precipitation (Serrano et al. 1999) and long-term trends in European air quality (Colette et al., 2016; Torseth et al., 2012). The main advantages, as discussed in the literature of the MK approach over linear regression for trend assessments are that (i) it does not require normally distributed data, (ii) it is not affected by outliers, and (iii) it removes the effect of temporal auto-correlation in the data (e.g., Gurreiro et al., 2014, Li et al., 2016, Torseth et al., 2012., Yao et al., 2016). Trend assessment using the LR approach have however been used in UK air quality monitoring network reports (e.g. Conolly et al., 2016); therefore both LR and MK approaches were used in this paper primarily as a quality assurance check.

**Author Response:** The authors have already responded to a similar comment raised by Reviewer 1 previously. To reiterate, the measured annual concentrations of $NH_3$ and $NH_4^+$ were shown as coloured dots on the map to illustrate the spatial variability across the UK. Interpolated concentration maps have not been used since the interpolation of the discrete measurement points (e.g. using bilinear interpolation) will give the reader a false sense of the spatial variability of air concentrations from the limited number of measurement locations.

The authors feel that these figures are relatively easily to compare by eye, but have added the sentence (highlighted) in the Figure 4 caption below to help draw the reader's eye:

**Figure 4: Measured annual mean concentrations from the UK National Ammonia Monitoring Network (NAMN) for 2005 for (a) $NH_3$ and (b) particulate $NH_4^+$, and maps at 5 km by 5 km grid resolution for 2005 of (c) the estimated annual $NH_3$ emissions (Dragosits et al. 2005) and (d) the dominant $NH_3$ emission source category (based on Hellsten et al., 2008), indicating the relationships between measured air concentrations and spatial variability in $NH_3$ sources emissions. The measurements show a broad pattern of small air concentrations across NW Scotland. Conversely, the largest concentrations occur in areas with intensive cattle, pig and poultry farming with high $NH_3$ emissions e.g. East Anglia in SE England.**

8) *"Page 9, lines 7-9 "The limited variation across the UK for the annual average NH4+ concentrations can be attributed to the atmospheric formation process (providing a diffuse source) and its longer atmospheric lifetime" This is very speculative, although the argument is one of potential causes."*

**Author Response:** The authors have replaced the above sentence with **"Particulate $NH_4^+$ is a slowly formed secondary product with a longer residence time in the atmosphere and is thus expected to show less spatial variation than $NH_3$. This is confirmed in the measurement data."**

9) *"Page 10, line 10, why use "for an example year of 2012s" but not 2005 as used early?"*

**Author Response:** The 2012 comparison (modelled vs monitored) updates an earlier inter-comparison assessment carried out for the year 2002 by Dore et al. (2007).

In section 3.1, the 2005 dominant sources map was used to classify NAMN sites according to one of seven dominant emission source categories. 2005 annual mean concentrations from the NAMN were therefore shown alongside annual $NH_3$ emissions and the dominant $NH_3$ emission source category for 2005 in Figure 4.

10) *"Page 10, lines 13-15 "The scatter may be explained by the large local spatial variability of NH3, related primarily to rapid decreases of NH3 concentrations with distance from a source (see e.g. Pitcairn et al., 1998; Dragosits et al., 2002),"* How come the references in ten year ago can support the interpretation of the difference in 2012? Is it still true and any evidence to say this?

**Author Response:** The two early references were quoted for the following reasons:

Pitcairn et al., (1998) is an early reference that measured the gradient in $NH_3$ concentrations downwind of an emission source. Concentrations close to the livestock buildings (within 300 m) were found to be very large, with concentrations reaching background concentrations at distances of about $1 - 2$ km from source.

Dragosits et al., (2002) presented a detailed assessment of the sub-grid spatial variability in $NH_3$ concentrations and deposition within a 5 km x 5 km case study site.

Large spatial variability of $NH_3$ at local, regional and regional scales is widely described in the literature, e.g. Li et al., 2017.

Li, Y., Thompson, T, M., Van Damme, M., Chen, X., Benedict, K. B., Shao, Y., Day, D., Boris, A., Sullivan, A. P., Ham, J., Whitburn, S., Clarisse, L., Coheur, P-F., and Collett Jr, J.L.: Temporal and spatial variability of ammonia in urban and agricultural regions of northern Colorado, United States. Atmospheric Chemistry and Physics, 17, 6197–6213, doi:10.5194/acp-17-6197-2017, 2017.

11) *"Page 10, lines 10-29, the agreement is of course more important than the correlation and should be discussed."*

**Author Response:** We think the reviewer is referring to the two paragraphs below (Page 10, lines 10-29): "Comparison of measurements with modelled $NH_3$ concentrations from the FRAME model for an example year of 2012 showed significant scatter when considering the full network of sites ($n = 85$, $R^2 = 0.62$) (Figure 5a). In this graph, each point is colour-coded according to the estimated dominant $NH_3$ emission source category for the 5 km by 5 km grid square. This updates a similar comparison from Sutton et al. (2001b) for the year 2000. The scatter may be explained by the large local spatial variability of $NH_3$, related primarily to rapid decreases of $NH_3$ concentrations with distance from a source (see e.g. Pitcairn et al., 1998; Dragosits et al., 2002), with the result that a single site measurement only gives an approximate indication of concentrations across the model grid square it is located in. At many of the sites where the model overestimates concentrations, the measurements are in fact carried out in nature reserves, or in clearings inside forests. The monitoring sites in these sink areas are typically well away from local sources. Conversely, some of the outliers where measurements are larger than the model predictions show indications of being affected by nearby emission sources, as was established by investigations during site visits.

Figure 6 considers measured $NH_3$ concentrations at a subset of sites (44 out of the full 85 sites) that are located away from nearby local sources, in forest or semi-natural areas, following the site classification and assessment by Hallsworth et al. (2010). For this restricted set of sites, $R^2 = 0.76$ for 2012 which is higher than the correlation for the overall UK network. The improvement in correlation between measured and modelled $NH_3$ concentrations for this subset of sites can be explained by the monitoring locations typically being further away from sources, so that uncertainties in local emissions estimates are to some extent averaged out. This observation is also consistent with the findings of Vieno et al. (2009)."

The authors have revised the two paragraphs to include more explanatory details and discussion – see below:

"The comparison of NAMN $NH_3$ and $NH_4^+$ measurements with modelled $NH_3$ concentrations from the FRAME model in this paper is made for an example year of 2012. This updates an earlier inter-comparison assessment carried out by Dore et al. (2007) for the year 2002, In the comparison of the FRAME model estimates (based on 2012 UK AENEID $NH_3$ emission data) with the NAMN measurement results for 2012 (Figure 5), the network annual mean concentrations for each site is compared against the model estimate for the 5-km grid square in which it occurs. Each point is also colour-coded according to the estimated dominant $NH_3$ emission source category for the 5 km by 5 km grid square, following the methodology described in a similar comparison from Sutton et al. (2001b) for the year 2000.

For $NH_3$, both the model estimates and the measurement agree that background and sheep sites are characterised by small $NH_3$ concentrations (< 1 µg $NH_3$ $m^{-3}$ annual mean), while agricultural areas, particularly areas with intensive pig and poultry areas, are associated with large $NH_3$ concentrations (up to 8 µg $NH_3$ $m^{-3}$ annual mean). Overall, the comparison suggests a fairly good fit with regard to both the magnitude and spatial variability of $NH_3$ concentrations at a national scale ($n = 85$), with an $R^2$ value of 0.6 (**Error! Reference source not found.**5a). UK $NH_3$ emissions with a 5×5 km grid-square resolution is used as input in the FRAME model and the accuracy of the emissions data is critical to the model performance. The broad agreement between measurement and FRAME estimates broadly support the predictions of the FRAME model, lending support to the AENEID model outputs. There are however significant scatter in the comparison, with some systematic differences in the comparison of FRAME and the measurements, depending on the air concentration and dominant source.

$NH_3$ is known to exhibit large sub-grid variability (e.g. Dragosits et al. 2002), influenced by proximity to emission source strength and type. In the vicinity of emission sources, $NH_3$ concentrations generally decay exponentially with distance away from source due to dispersion and dilution (e.g. Pitcairn et al., 1998). As it is a highly reactive gas, a significant fraction of the $NH_3$ emitted is also rapidly deposited within 1 km radius of the source, so that concentrations reach background concentrations at distances of about $1 - 2$ km from source (Fowler et al., 1998). This effect is particularly important in areas with high local variability in $NH_3$ emissions, such as intensive agricultural areas. The observed scatter in the comparison may therefore be due to the spatial location of the sampling site relative to the distribution of sources. For example, at many of the sites where the model overestimates concentrations, the measurements are in fact made in nature reserves, or in clearings inside forests. The monitoring sites in these sink areas are typically well away from local sources and which would on average be more distant from sources than assumed in the FRAME 5 km average estimates, thereby underestimating concentrations, Conversely, some of the outliers where measurements are larger than the model predictions show indications of being affected by nearby emission sources, as was established by investigations during site visits. This effect is particularly important in areas with high local variability in ammonia emissions, such as intensive agricultural areas and illustrates the importance of having a large number of sites for comparison.

Figure 6 considers measured $NH_3$ concentrations at a subset of sites (44 out of the full 85 sites) that are located away from nearby local sources, in forest or semi-natural areas, following the site classification and assessment by Hallsworth et al. (2010). For this restricted set of sites, $R^2 = 0.76$ for 2012 which is higher than the correlation for the overall UK network. The improvement in correlation between measured and modelled $NH_3$ concentrations for this subset of sites can be explained by the monitoring locations typically being further away from sources, so that uncertainties in local emissions estimates are to some extent averaged out. This observation is also consistent with the findings of Vieno et al. (2009)."

Dore, A. J., Vieno, M., Tang, Y. S., Dragosits, U., Dosio, A., Weston, K. J., & Sutton, M. A. (2007). Modelling the atmospheric transport and deposition of sulphur and nitrogen over the United Kingdom and

assessment of the influence of SO 2 emissions from international shipping. *Atmospheric Environment*, *41*(11), 2355-2367.

Dore, A. J., Carslaw, D. C., Braban, C., Cain, M., Chemel, C., Conolly, C. & Lawrence, S. (2015). Evaluation of the performance of different atmospheric chemical transport models and inter-comparison of nitrogen and sulphur deposition estimates for the UK. *Atmospheric Environment*, *119*, 131-143.

Sutton, M. A., Tang, Y. S., Dragosits, U., Fournier, N., Dore, A. J., Smith, R. I., Weston, K. J., and Fowler, D.: A spatial analysis of atmospheric ammonia and ammonium in the U.K, ScientificWorldJournal, 1 Suppl 2, 275-286, 10.1100/tsw.2001.313, 2001b.

12) *"Page 10, lines 39-41 "This suggests either too low a formation rate for NH4+ in the model at cleaner sites, or too high a removal rate for NH4+, or a combination of both. The presence of higher measured NH4+ concentrations in remote areas than shown by the model may also indicate that NH4+ has a longer residence time than treated in the model." This is very speculative. Under prediction of particulate NH4+ by air quality models is very common in the remote clean atmosphere due to those simple assumptions. "*

**Author Response:** We agree that models may underestimate particulate $NH_4^+$ due to simplistic assumptions. The models uses simple chemistry schemes and deposition velocities from the literature. Our text already includes the implication that the model treatment may have shortcomings.

13) *"Page 11, lines 19-24, all arguments are absolutely correct, but then what? In other words, how these directly explain the seasonal variation?"*

**Author Response:** The authors feel that the seasonal variations in $NH_3$ and $NH_4^+$ have been fully explained and discussed fully in <Section 3.3. Seasonal variability in measured UK $NH_3$ and $NH_4^+$ concentrations>, and that there is nothing more usefully to add to this section.

14) *"Page 11, lines 28-29 "A smaller peak in NH3 can also be seen annually in April, which indicates potential longer range influences of manure spreading in spring, even at this remote location (Figure 8b)." The argument is also very speculative. Solid evidence is needed."*

**Author Response:** The application of manure and fertilizers in the UK occurs predominantly during spring, but to a lesser extent in autumn farmlands and agriculture. Peak $NH_3$ concentrations are often measured in April at sites within agricultural landscapes. It is thus perfectly plausible that manure spreading activities within or outside the 5 km x 5 km grid square containing the background site shows up as a small April peak, depending on which way the wind is blowing.

 *"Page 12, lines 5-6, "Interestingly, the dip in concentrations in June matches a period when crops will be actively growing with possible uptake and removal of NH3 from the atmosphere." Evidence but not just arguments!"*

**Author Response:** Additional text and references (highlighted) added – see below:

"Interestingly, the dip in concentrations in June matches a period when crops will be actively growing with possible uptake and removal of $NH_3$ from the atmosphere. Vegetation can be a source or a sink of atmospheric $NH_3$ and uptake of $NH_3$ can occur when the relative concentration of $NH_3$ in the atmosphere is higher than inside the plant stoma (e.g. Sutton et al.,1995, Massad et al., 2010; Flechard et al., 2013).

Flechard, C. R., Massad, R. S., Loubet, B., Personne, E., Simpson, D., Bash, J. O., Cooter, E. J., Nemitz, E., and Sutton, M. A.: Advances in understanding, models and parameterizations of biosphere-atmosphere ammonia exchange, Biogeosciences, 10, 5183-5225, 10.5194/bg-10-5183-2013, 2013.

Massad, R-S., Nemitz, E. and Sutton, M.A.: Review and parameterisation of bi-directional ammonia exchange between vegetation and the atmosphere, Atmospheric Chemistry and Physics, 10, 10359–10386, doi:10.5194/acp-10-10359-2010, 2010

Sutton, M. A., Schjoerring, J. K., and Wyers, G. P.: Plant-atmosphere exchange of ammonia, Philos. T. Roy. Soc. S-A, 351,261–278, 1995.

16) *"Page 12, lines 10-13 "Although the formation of particulate NH4+ primarily depends on the occurrence of NH3 in the atmosphere, synoptic meteorology and long range transboundary transport from continental Europe are important drivers influencing the seasonal variations of NH4+ across the UK, due to its' longer lifetime." The part is totally confused and which studies support these?"*

**Author Response:** the references which supports the statement quoted is added at the end of the statement – see below:

"Although the formation of particulate $NH_4^+$ primarily depends on the occurrence of $NH_3$ in the atmosphere, synoptic meteorology and long range transboundary transport from continental Europe are important drivers influencing the seasonal variations of $NH_4^+$ across the UK, due to its longer lifetime (Vieno et al., 2014, 2016)."

Vieno, M., Heal, M. R., Hallsworth, S., Famulari, D., Doherty, R. M., Dore, A. J., Tang, Y. S., Braban, C. F., Leaver, D., Sutton, M. A., and Reis, S.: The role of long-range transport and domestic emissions in determining atmospheric secondary inorganic particle concentrations across the UK, Atmospheric Chemistry and Physics, 14, 8435-8447, 10.5194/acp-14-8435-2014, 2014.

Vieno, M., Heal, M. R.,Williams, M.L., Carnell, E.J., Nemitz, E., Stedman, J.R., and Reis, S.: The sensitivities of emissions reductions for the mitigation of UK $PM_{2.5}$ , Atmospheric Chemistry and Physics, 16, 265-276, doi: 10.5194/acp-16-265-2016, 2016.

17) *"Page 12, lines 10-35, it is odd. Why not use modeling results,.e.g., processing analysis, to interpret? It is very straight forward by comparing with current arguments."*

**Author Response:** The authors think the reviewer is referring to the text below. In which case the authors are not entirely clear what the reviewer is asking. We reiterate that this paper is an analysis of what can be learnt from trends and patterns in measurement data. As has already been noted above, it is accepted that modelling of these processes can have limitations.

"For particulate $NH_4^+$, as expected for a secondary pollutant, concentrations are more decoupled from the dominant $NH_3$ source sectors in the vicinity of a site. Although the formation of particulate $NH_4^+$ primarily depends on the occurrence of $NH_3$ in the atmosphere, synoptic meteorology and long range transboundary transport from continental Europe are important drivers influencing the seasonal variations of $NH_4^+$ across the UK, due to its' longer lifetime. The seasonal trends in particulate $NH_4^+$ are seen to be broadly similar for the four different emission source sectors (Figure 7b), with the magnitude of the $NH_4^+$ concentrations reflecting $NH_3$ concentrations at a regional level. In the atmosphere, particulate $NH_4^+$ are primarily in the form of $(NH_4)_2SO_4$ and $NH_4NO_3$, formed when the acid gases $HNO_3$ and $H_2SO_4$.in the atmosphere are neutralised by $NH_3$ (Putaud et al., 2010). $NH_3$ preferentially neutralizes $H_2SO_4$ due to its low saturation vapour pressure (forming $NH_4HSO_4$ then $(NH_4)_2SO_4$), while $NH_4NO_3$ is formed when abundant $NH_3$ is available, In contrast to $(NH_4)_2SO_4$, $NH_4NO_3$ is a semi-volatile component (Stelson & Seinfeid, 1982). Long-term data from the UK Acid Gas and Aerosol Network (AGANet, Conolly et al., 2016) shows a change in the particulate phase of $NH_4^+$ from $(NH_4)_2SO_4$ to $NH_4NO_3$, with particulate nitrate concentrations exceeding that of particulate sulphate approximately three-fold (on a molar basis) (Fig. 18a). This suggests that the thermodynamic equilibrium between the gas phase $NH_3$ and $HNO_3$ and the aerosol phase $NH_4NO_3$ will have a much greater effect on the seasonal concentrations of $NH_4^+$ than $(NH_4)_2SO_4$. The formation and dissociation of $NH_4NO_3$ depend strongly on ambient temperature and humidity (Stelson and Seinfeld, 1982). Warm, dry weather in summer promotes dissociation, decreasing particulate phase $NH_4NO_3$ relative to gas phase $NH_3$ and $HNO_3$. During the winter months, low temperature and high humidity favour the formation of $NH_4NO_3$ from the gas phase $NH_3$ and $HNO_3$. By contrast, the spring peak in $NH_4^+$ concentrations may be attributed to photochemical processes (elevated ozone) leading to enhanced formation of $HNO_3$ during this period (Pope et al., 2016) and also to import of particulate $NO_3^-$ through long-range transboundary transport, e.g. from continental Europe, as discussed in Vieno et al. (2014). Nevertheless, it is notable that the winter minima for $NH_4^+$ aerosol concentrations at sheep and background sites are more pronounced than that for pig, poultry and cattle dominated sites. This may be a result of a combination of smaller $NH_3$ emissions in winter in these areas (as indicated by Figure 7a) and differences in long-range transport to the more remote areas in winter conditions."

18) *"The analysis in Section 3.5.1 does not sound scientific when analytic errors were considered. The same concern is applicable for Section 3.5.4."*

**Author Response:** It is not clear what the reviewer means by *",,,does not sound scientific when analytic errors were considered "*

<*Section 3.5.1*. Mann-Kendall non-parametric time series analysis> The MK approach was applied to the long-term dataset using the methodology described. It is a commonly used approach, as acknowledged by the reviewer.

<Section 3.5.4. Influence of climate> An exponential model (using R) was used to fit the data (NH3 vs temperature/rainfall) and to determine the significance of the regression.

> 19) *"Section 3.5.6, a bunch of arguments are redundant.Technically, the study is informative and valuable. However, this reviewer believes that a few cavities are still there. It should be improved to better service research community."*

**Author Response:** <Section 3.5.6. Changing chemical climate and effects on long-term trends in $NH_3$ and $NH_4^+$> We feel that sufficient evidence have been presented to support the findings in this section. As the reviewer has not specified which arguments they find redundant, the authors have to disagree with the reviewer and look to the editor to adjudicate.

Previous communication regarding reviewer 2 comments:

*Author query: " the reviewer does not discuss what was wrong with our response to the first review in any quantitative way, while re-iterating some of the same unsubstantiated issues from the first review, e.g. "authors failed to demonstrate why the selected statistical analysis too can address..." where we fully referenced the methods we used."*

*Reviewer response: "The reviewer's second round comments are based on the revised version, which no track changes or highlights can be found for revision.*

**Author response:** The track changes version was uploaded and is available in the manuscript management page.

*Reviewer response: A bunch of statistical analysis tools are available for trend analysis, but the reviewer cannot find any scientific reason in the revised version for the authors' selection. The reviewer fully agrees that the authors employed popular ones which have been widely used in literature. If the authors really want to emphasize so-called "standard" (the reviewer would like to believe the "standard" should be updated with improving knowledge), their description and citation technically sound. However, it does not mean the selected tools are scientifically applicable for their specific datasets, at least to this reviewer and his experience on applying these tools.*

**Author response:** Given that reviewer 2 acknowledges that the methods used are popular in the peer review literature i.e. are a proven method for trend analysis scientifically, it is not inappropriate to use them. There may be many other statistical methods available, we are not sure which one is the Reviewer's preferred method, however all data in this paper are available for others to analysis with whichever statistical analysis they wish. We do not think we need to re-analyse for this paper as we have used valid techniques.

*Reviewer response: The justification for choosing the tools by the authors, particularly for studying the long-term trend in atmospheric ammonia, is very important and helpful for potential readers. I hope that the authors can agree and lead a way on this issue.*

**Author response:** We added in the revised manuscript already the justification of the two methods both scientifically plus noting that the UK and European assessments use these tools so we are consistent.

*Author query 2: the reviewer writes " A few comments are listed, but not just limited these." I would rather the reviewer detail all their queries so that they can be addressed rather than iterate through the review process again.*

*Reviewer response: Those minor issues haven't been listed. The reviewer fully believes that the authors can find technically. Again, the reviewer has no intention to delay the study for publishing if these cavities listed can be properly fixed.*

*Author query 3: The reviewer is very vague in some of the comments, e.g. "Section 3.5.6, a bunch of arguments are redundant": I am not sure we can respond to such a non-specific comment.*

*Reviewer response: The reviewer believes that the authors have provided sufficient evidences at the last few paragraphs in Section 3.5.6. A bunch of redundant arguments could set a bad example for young students. I hope that the authors can agree on this.*

**Author response:** As the reviewer themselves states, we have provided sufficient evidence, rather than having redundant arguments, and we fully believe the paper is an example of discussing the facts and issues and scientific questions. As the reviewer has not specified which arguments they find redundant, the authors just have to disagree with the reviewer and look to the editor to adjudicate.

To note, we responded in detail to Reviewer 1 and they were happy with the revision we provided.

*Reviewer response: all scientists have to be forced by the so-called "standard", the reviewer concerns where new science comes from.*

**Author response:** Reviewer 1 provided a detailed and thorough review. New science can only come through clear communication, which Reviewer 2 is not providing a good example of. Before we respond to Reviewer 2, which we are happy to do scientifically, could you check to see if the standard of the review are up to ACPD standards and advise.

*Reviewer response: Again, the reviewer fully believes the study can be improved better.*

Revised manuscript with all track changes (including revision 1 after first review):

[revised manuscript text omitted]